# Analytical expression for $\pi$-ton vertex contributions to the optical conductivity

**Juraj Krsnik,[1,2]⋆ Anna Kauch,[1] and Karsten Held[1]**

**1** Institute of Solid State Physics, TU Wien, 1040 Vienna, Austria
**2** Department for Research of Materials under Extreme Conditions,
Institute of Physics, HR-10000 Zagreb, Croatia

⋆ jkrsnik@ifs.hr

## Abstract

Vertex corrections from the transversal particle-hole channel, so-called $\pi$-tons, are generic in models for strongly correlated electron systems and can lead to a displaced Drude peak (DDP). Here, we derive the analytical expression for these $\pi$-tons, and how they affect the optical conductivity as a function of correlation length $\xi$, fermion lifetime $\tau$, temperature $T$, and coupling strength to spin or charge fluctuations $g$. In particular, for $T \rightarrow T_c$, the critical temperature for antiferromagnetic or charge ordering, the dc vertex correction is algebraic $\sigma_{VERT}^{dc} \propto \xi \sim (T - T_c)^{-\nu}$ in one dimension and logarithmic $\sigma_{VERT}^{dc} \propto \ln \xi \sim \nu \ln(T - T_c)$ in two dimensions. Here, $\nu$ is the critical exponent for the correlation length. If we have the exponential scaling $\xi \sim e^{1/T}$ of an ideal two-dimensional system, the DDP becomes more pronounced with increasing $T$ but fades away at low temperatures where only a broadening of the Drude peak remains, as it is observed experimentally, with the dc resistivity exhibiting a linear $T$ dependence at low temperatures. Further, we find the maximum of the DPP to be given by the inverse lifetime: $\omega_{DDP} \sim 1/\tau$. These characteristic dependencies can guide experiments to evidence $\pi$-tons in actual materials.



# 1   Introduction

The phenomenon of a displaced Drude peak (DDP) in metallic systems, characterized by a maximum in the optical absorption at a finite frequency (unlike in normal metals where the maximum occurs at zero frequency), has been observed over the past few decades across a diverse range of compounds including cuprates, transition metal oxides, organic conductors, and Kagome metals [1–23]. Despite the variety of these materials, a universal experimental feature is that the DDP frequency is an increasing function of temperature, $\omega_{DDP} \sim T^{\alpha}$, with the coefficient $\alpha$ in the range $0 < \alpha < 3/2$ [24]. This striking universal temperature dependence immediately raises an important question: Is there a microscopic mechanism common to all these materials that gives rise to a displaced Drude peak? In addressing this question, it is important to note the key similarities that these materials share: they are predominantly strongly correlated electron systems, many showing effectively two-dimensional (2D) physics and hosting strong spin and/or charge fluctuations.

Although several theories have been proposed in the past decade to explain the mechanism behind the Drude peak displacement on a broader level [24–27], our understanding of the phenomenon remains relatively limited. For example, Ref. [25] explains it in terms of the hydrodynamics of short-range quantum critical fluctuations of incommensurate density wave order. Another established scenario involves the transient localization mechanism [27], which originates from quantum localization corrections due to slow phononic fluctuations [24] or charge fluctuations mediated by long-range Coulomb interaction coexisting with lattice frustration [26] in low-dimensional systems. More recently, however, a novel mechanism involving $\pi$-ton vertex contributions [28–31] has been identified as another potential cause of the Drude peak displacement [32].

The significance of $\pi$-ton vertex contributions in shaping the optical spectrum of 2D correlated electron systems was first emphasized in Refs. [28–30] for a variety of different models of strongly correlated electron systems. These works provide a comprehensive analysis of different vertex contributions in correlated electron systems based on the two-particle reducibility, which was possible due to recent methodological advances in using the parquet

equations [33–35] within the dynamical vertex approximation (DΓA) [36–38] and the parquet approximation [33]. In particular, it was observed that the dominant vertex contributions in prototypical models of strongly correlated electrons originate from the transversal particle-hole ($\overline{ph}$) channel. Despite the negligibly small transfer momentum of the photon, these vertex contributions can pick up bosonic fluctuations at an arbitrary wave vector, and in particular strong antiferromagnetic (AFM) or charge density wave fluctuations at $\mathbf{k} - \mathbf{k}' \approx (\pi, \pi, ...)$ associated with correlated systems, thus the name $\pi$-tons [29]. The aforementioned vertex contributions from quantum localization, in contrast, emerge from the particle-particle ($pp$) channel.

Shortly after the numerical papers [28–30], $\pi$-tons were investigated in the simplified random phase approximation (RPA) [31,32,39,40]. These studies sought to better understand the behavior of $\pi$-ton vertex contributions in the weakly correlated regime of the Hubbard model across a broader temperature range, particularly near the paramagnetic-to-antiferromagnetic transition boundary. While it was first reported in Ref. [31] that the RPA $\pi$-ton vertex contributions are small compared to the bubble contribution in the 2D case, in Refs. [39,40] their nonnegligible contributions were recognized in one-dimensional (1D) systems until finally it was realized in Ref. [32] that they may lead to the DDP. Specifically, it was shown that, in the 1D case close to the paramagnetic-to-antiferromagnetic transition, the coupling of strong AFM fluctuations via the RPA $\pi$-ton vertex contributions to low-energy quasiparticle excitations shifts the Drude peak to a finite frequency [32]. Similar qualitative features were observed in the 2D case, but the magnitude of the $\pi$-ton vertex contributions was orders of magnitude smaller than in 1D, resulting in only a broadening of the Drude peak. It is important to note, however, that the study in Ref. [32] was purely numerical, and achieving convergence of the $\pi$-ton vertex contributions in the 2D case near the transition boundary proved to be a formidable task.

Building on these findings, in this paper, we further investigate the impact of $\pi$-tons in 2D systems with strong AFM fluctuations by conducting an analytical evaluation of the $\pi$-ton vertex contributions and by leveraging the cuba package [41] for adaptive integration. The analytical approach allows us to examine $\pi$-ton vertex contributions arbitrarily close to the transition boundary, while the improved adaptive integration (compared to Ref. [32]) enables us to benchmark our analytical results over a broader temperature range. To accomplish this, we are taking only the basic ingredients required to obtain the DDP via $\pi$-tons as noted in Ref. [32]: (i) low-energy fermionic quasiparticle excitations described by Green's function [42]

$$G(\mathbf{k}, i\nu_m) = \frac{1}{i\nu_m - \varepsilon_{\mathbf{k}} + \frac{i}{2\tau}\mathrm{sgn}\,\nu_m}, \tag{1}$$

and (ii) AFM fluctuations resembling the Ornstein-Zernike form [43–45]

$$\chi_{\mathrm{OZ}}(\mathbf{q}, i\omega_m) \sim \frac{A}{\xi^{-2} + (\mathbf{q} - \mathbf{Q})^2 + \lambda\,|\omega_m|}, \tag{2}$$

with $\mathbf{Q} = (\pi, \pi, ...)$. Moreover, we assume a D-dimensional half-filled hypercubic lattice with $N$ sites and only the nearest-neighbor hoppings $t$, for which the electron dispersion equals $\varepsilon_{\mathbf{k}} = -2t\sum_{i=1}^{D}\cos(k_i)$. We set $t \equiv 1$ as the unit of energy, as well as $\hbar \equiv 1$, $k_B \equiv 1$, electric charge $e \equiv 1$, and lattice constant $a \equiv 1$. For the fermion lifetime $\tau$, we assume the quadratic-in-temperature Fermi liquid form [42] $\tau^{-1} \sim a + bT^2$, where the constant term may originate from impurity or some other source of scattering.

With $G(\mathbf{k}, i\nu_m)$ fully specified, we can further evaluate the Lindhard function and with it the RPA $\pi$-ton vertex function, assuming a Hubbard repulsion $U \leq 2$ between fermions [31]. We then seek the paramagnetic-to-antiferromagnetic transition temperature $T_c$ by identifying the temperature at which the vertex function (proportional to magnetic susceptibility) diverges. Finally, for temperatures $T > T_c$, we can fit the vertex function calculated in RPA

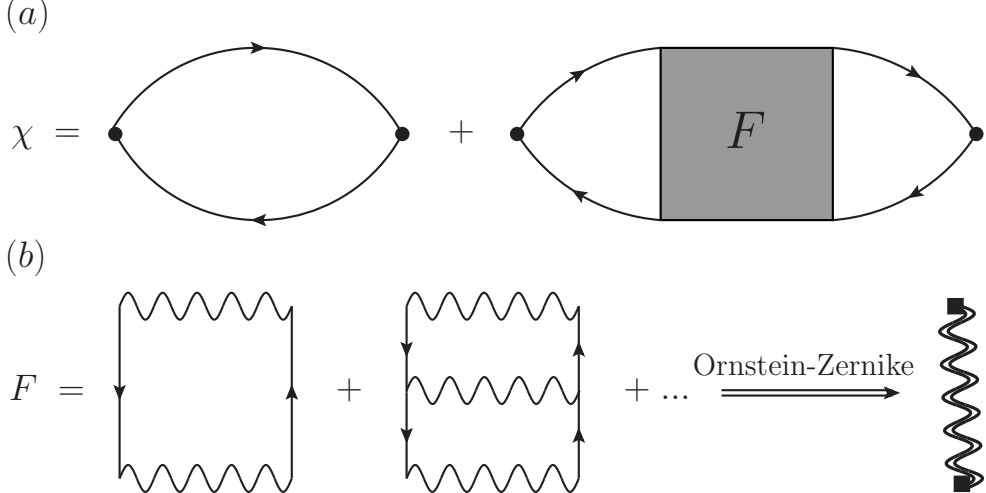

Fig. 1: (a) Diagrammatic representation of the current-current correlation function $\chi_{jj}$ with bubble (left) and vertex contribution (right). (b) Diagrammatic representation of the $\pi$-ton vertex corrections in the $\overline{ph}$ channel within the RPA. Here, the solid lines represent the fermion Green's functions $G$, the wavy lines the Hubbard interaction $U$, $F$ is the vertex function, while light-fermion vertices are denoted by solid circles. To facilitate analytical evaluation, we approximate the RPA vertex function using the Ornstein-Zernike form. In this approximation, the vertex function adopts the form of an overdamped boson (double wavy line). The full squares represent interaction vertices between fermions and overdamped antiferromagnetic spin fluctuations.

to the Ornstein-Zernike form for obtaining the temperature dependence of the parameters $\xi$, $A$, and $\lambda$, giving us all the necessary quantities to evaluate $\pi$-ton vertex contributions to the optical conductivity. This whole procedure for $U = 2$ in the 1D case and for $U = 1.9$ in the 2D case with $\tau^{-1} = 0.1547 + 1.637\,T^2$ has been already carried out in Refs. [32] and [31], respectively. Here, we instead use these fitted temperature dependencies and substitute them in the analytical expression for $\pi$-tons to (i) perform benchmark calculations against numerical results, and (ii) investigate scenarios with the Drude peak displacement in 2D. Please note that while our present consideration specifically considers AFM fluctuations in the $\pi$-ton vertex contributions, these fluctuations can actually be of any origin, e.g., also stem from charge fluctuations, provided they are well described by the Ornstein-Zernike form in Eq. (2).

The paper is structured as follows: In Sec. 2.1, we recall the evaluation of the Drude optical conductivity from the bubble contribution, prior to the evaluation of the $\pi$-ton vertex contributions in Sec. 2.2. In Sec. 3, we first discuss general qualitative features of the analytically obtained $\pi$-ton vertex contributions, after which in Sec. 3.1 we benchmark our analytical results with the results of the adaptive integration in 1D and 2D. We present our key findings on the potential pathways to the DDP in the presence of $\pi$-tons in 2D systems in Sec. 3.2. Lastly, we compare the $\pi$-ton vertex contributions to the (quantum) localization vertex contributions in Sec. 3.3, before concluding our findings in Sec. 4.

## 2 Analytical evaluation of the optical conductivity

### 2.1 Bubble contribution

Since similar concepts will be applied in calculating the $\pi$-ton vertex contributions, we first recall the textbook derivation [42] of the Drude optical conductivity [46,47] from the bubble (BUB) contribution to the current-current correlation function, the leftmost diagram in Fig. 1(a). In terms of the fermion Green's functions $G(\mathbf{k}, i\nu_m)$, fermion velocity $\nu_\mathbf{k} = \frac{\partial \varepsilon_\mathbf{k}}{\partial \mathbf{k}}$ [48], and temperature $T(\beta^{-1})$, the latter can be expressed in the long-wavelength limit $\mathbf{q} \to 0$ as [31]

$$
\chi_{BUB}(i\omega_n) = -\frac{2}{\beta N} \sum_{i\nu_m} \sum_\mathbf{k} \nu_\mathbf{k}^2 G(\mathbf{k}, i\nu_m) G(\mathbf{k}, i\nu_m + i\omega_n) \tag{3}
$$

$$
= -\frac{2}{\beta N} \sum_{i\nu_m} \sum_\mathbf{k} \nu_\mathbf{k}^2 \left[ G^e(\mathbf{k}, i\nu_m) + G^h(\mathbf{k}, i\nu_m) \right] \left[ G^e(\mathbf{k}, i\nu_m + i\omega_n) + G^h(\mathbf{k}, i\nu_m + i\omega_n) \right].
$$

Here, we introduced $G^{e/h}(\mathbf{k}, i\nu_m) \equiv \frac{1}{i\nu_m - \varepsilon_\mathbf{k} \pm \frac{i}{2\tau}} \Theta(\pm \nu_m)$ for the Green's function at $\nu_m > 0$ and $\nu_m < 0$, respectively. These can be associated with the propagation of an electron/hole, and the continued $G^{e/h}(\mathbf{k}, i\nu_m)$ has poles in the lower/upper half of the complex plane. In terms of $G^{e/h}$, $G$ in Eq. (1) can be simply rewritten as $G = G^e + G^h$.

In the following, we consider $\omega_n \geq 0$ since the bosonic correlation function is symmetric in frequency. This implies for $\nu_m$: $\omega_n > -\nu_m > 0$, since otherwise two electrons or two holes would be created by an incident photon. Mathematically, the two contributions $G^h(\mathbf{k}, i\nu_m) G^h(\mathbf{k}, i\nu_m + i\omega_n)$ and $G^e(\mathbf{k}, i\nu_m) G^e(\mathbf{k}, i\nu_m + i\omega_n)$ have all their poles located in the same half of the complex plane so they contribute zero, while $G^e(\mathbf{k}, i\nu_m) G^h(\mathbf{k}, i\nu_m + i\omega_n)$ cannot be realized for $\omega_n > 0$. That is, among the four terms in Eq. (3) the only one that has a non-zero contribution is $G^h(\mathbf{k}, i\nu_m) G^e(\mathbf{k}, i\nu_m + i\omega_n)$, where $G^h(\mathbf{k}, i\nu_m)$ and $G^e(\mathbf{k}, i\nu_m + i\omega_n)$ describe the propagation of a hole and an electron, respectively, excited by the incident photon, as depicted on the left-hand side of Fig. 2.

Fermions in the vicinity of the Fermi surface (FS) contribute the most to the current fluctuations. For that reason, we replace the momentum summation in Eq. (3) by an energy integral in which the fermion density of states $g$ is approximated by a constant, i.e., the value at the Fermi level $g(\varepsilon_F)$ [42]

$$
\frac{1}{N} \sum_\mathbf{k} \nu_\mathbf{k}^2 u_\mathbf{k} = \int_{-\infty}^{+\infty} \nu^2(\varepsilon) g(\varepsilon) u(\varepsilon) d\varepsilon \approx g(\varepsilon_F) \langle \nu^2 \rangle_{FS} \int_{-\infty}^{+\infty} u(\varepsilon) d\varepsilon, \tag{4}
$$

with $\langle \nu^2 \rangle_{FS}$ being the fermion velocity averaged over the Fermi surface, holding for an arbitrary function $u_\mathbf{k}$. This further gives, see Appendix A for details on the integral evaluation:

$$
\chi_{BUB}(i\omega_n) = -\frac{2}{\beta} g(\varepsilon_F) \langle \nu^2 \rangle_{FS} \sum_{\omega_n > -\nu_m > 0} \int_{-\infty}^{+\infty} d\varepsilon \left[ \frac{1}{i\omega_n + i\nu_m - \varepsilon + \frac{i}{2\tau}} \right] \left[ \frac{1}{i\nu_m - \varepsilon - \frac{i}{2\tau}} \right]
$$

$$
= -\frac{2}{\beta} g(\varepsilon_F) \langle \nu^2 \rangle_{FS} \frac{2\pi i}{i\omega_n + \frac{i}{\tau}} \sum_{\omega_n > -\nu_m > 0} 1 = -2 g(\varepsilon_F) \langle \nu^2 \rangle_{FS} \frac{1}{i\omega_n + \frac{i}{\tau}} i \frac{2\pi n}{\beta} \tag{5}
$$

$$
= -2 g(\varepsilon_F) \langle \nu^2 \rangle_{FS} \frac{i\omega_n}{i\omega_n + \frac{i}{\tau}},
$$

where we note that the sum over $\nu_m$ in the second row gave the index $n$ of the Matsubara frequency $\omega_n$. Analytic continuation, $i\omega_n \to \omega + i0^+$, now readily yields

$$
\chi_{BUB}(\omega) = -2 g(\varepsilon_F) \langle \nu^2 \rangle_{FS} \frac{\omega}{\omega + \frac{i}{\tau}}, \quad \text{and} \quad \text{Im}\chi_{BUB}(\omega) = 2 g(\varepsilon_F) \langle \nu^2 \rangle_{FS} \tau \frac{\omega}{1 + \omega^2 \tau^2}. \tag{6}
$$

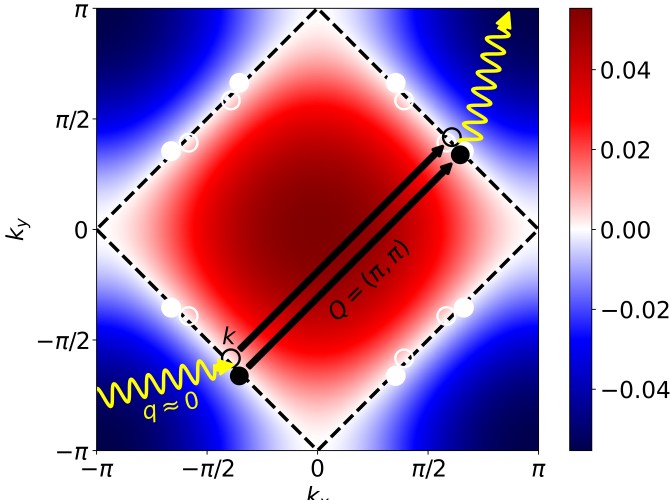

Fig. 2: Sketch of the dominant $\pi$-ton process for the case with weakly interacting fermions on a half-filled square lattice. The yellow wiggled line denotes the incoming (and outgoing) photon with the transfer momentum $\mathbf{q} \approx 0$, which excites an electron-hole pair represented by the full and empty black circles, respectively. The excited hole (electron) with wave vector $\mathbf{k}$ $(\mathbf{k}+\mathbf{q})$ is scattered via antiferromagnetic fluctuations with wave vector $\mathbf{Q} = (\pi, \pi)$ (black arrows) across the Fermi surface (black dashed line) forming a second electron-hole pair that eventually recombines to emit the outgoing photon. The full (empty) white circles show the equivalent eightfold symmetric electron-like (hole-like) states. The red (blue) color region denotes hole-like (electron-like) states.

Since the optical conductivity is given in terms of the current-current correlation function on the real frequency axis as $\sigma(\omega) = \frac{\text{Im}\chi(\omega)}{\omega}$ [31], we recover the Drude result

$$\sigma_{BUB}(\omega) = \frac{\sigma_{BUB}^{dc}}{1 + \omega^2 \tau^2}\,, \tag{7}$$

where we have identified $\sigma_{BUB}^{dc} \equiv 2g(\varepsilon_F)\langle v^2 \rangle_{FS} \tau$. For the practical purposes of this paper, we obtain $\sigma_{BUB}^{dc}$ by performing adaptive integration of the bubble contribution on the real frequency axis, as outlined in Ref. [31].

## 2.2 $\pi$-ton vertex contributions

The total vertex (VERT) contribution to the current-current correlation function illustrated by the rightmost diagram in Fig. 1(a) reads

$$\chi_{VERT}(\mathbf{q}, i\omega_n) = -\frac{2}{(\beta N)^2} \sum_{i\nu_m, i\nu_{m'}} \sum_{\mathbf{k},\mathbf{k}'} v_{\mathbf{k},\mathbf{q}} v_{\mathbf{k}',-\mathbf{q}} G(\mathbf{k}, i\nu_m) G(\mathbf{k}+\mathbf{q}, i\nu_m + i\omega_n)$$
$$\times G(\mathbf{k}', i\nu_{m'}) G(\mathbf{k}'+\mathbf{q}, i\nu_{m'} + i\omega_n) F_d(\mathbf{k}, \mathbf{k}', \mathbf{q}, i\nu_m, i\nu_{m'}, i\omega_n), \tag{8}$$

whose evaluation requires knowledge of the full density component of the two-particle vertex $F_d(\mathbf{k}, \mathbf{k}', \mathbf{q}, i\nu_m, i\nu_{m'}, i\omega_n)$ [35,49]. Assuming the predominance of the $\pi$-ton vertex contributions, it is, however, reasonable to approximate the entire density component using only the $\overline{ph}$ contribution $F_d(\mathbf{k}, \mathbf{k}', \mathbf{q}, i\nu_m, i\nu_{m'}, i\omega_n) \approx F_{d,\overline{ph}}(\mathbf{k}, \mathbf{k}', \mathbf{q}, i\nu_m, i\nu_{m'}, i\omega_n)$.

Within the RPA framework, the vertex function $F_{d,\overline{ph}}(\mathbf{k}, \mathbf{k}', \mathbf{q}, i\nu_m, i\nu_{m'}, i\omega_n)$ can be further simplified, as the RPA $\pi$-ton vertex function depends only on a single transfer momentum/energy, $F_{d,\overline{ph}}(\mathbf{k}, \mathbf{k}', \mathbf{q}, i\nu_m, i\nu_{m'}, i\omega_n) \equiv F_{\overline{ph}}(\mathbf{k}' - \mathbf{k}, i\nu_{m'} - \nu_m)$ [31, 32, 39, 40]. The corresponding RPA $\pi$-ton vertex function is depicted in Fig. 1(b). Further, close to the paramagnetic-to-antiferromagnetic phase transition, the $\pi$-ton vertex function can be well approximated with the Ornstein-Zernike form [31, 32] (diagrammatically represented by the double wavy line in Fig. 1(b))

$$F_{\overline{ph}}(\mathbf{k}' - \mathbf{k}, i\nu_{m'} - i\nu_m) \approx F_{OZ}(\mathbf{k}' - \mathbf{k}, i\nu_{m'} - i\nu_m) = \frac{A}{\xi^{-2} + (\mathbf{k}' - \mathbf{k} - \mathbf{Q})^2 + \lambda|\nu_{m'} - \nu_m|}, \quad (9)$$

with $\mathbf{Q} = (\pi, \pi, ...)$ corresponding to the strong AFM fluctuations.[1] Here, $\xi$ is the AFM correlation length, $\lambda$ represents the damping of AFM fluctuations, while $A \sim g^2$ contains the coupling strength $g$ (full squares in Fig. 1(b)) of fermions to AFM fluctuations. Considering the momentum and frequency characteristics of $F_{OZ}$, it is evident that the $\pi$-ton contributions are most significant when the energy transfer is approximately zero, i.e., $\nu_{m'} - \nu_m \approx 0$, and the momentum transfer is close to $\mathbf{Q}$, i.e., $\mathbf{k}' - \mathbf{k} \approx \mathbf{Q}$. Thus, it is convenient to rewrite Eq. (8) with a change of variables $\mathbf{k}' = \mathbf{k} + \mathbf{Q} + \tilde{\mathbf{q}}$, yielding

$$\chi_{VERT}(i\omega_n) = -\frac{2}{(\beta N)^2} \sum_{i\nu_m, i\nu_{m'}} \sum_{\mathbf{k}, \tilde{\mathbf{q}}} \nu_{\mathbf{k}} \nu_{\mathbf{k}+\mathbf{Q}+\tilde{\mathbf{q}}} G(\mathbf{k}, i\nu_m) G(\mathbf{k}, i\nu_m + i\omega_n)$$
$$\times G(\mathbf{k} + \mathbf{Q} + \tilde{\mathbf{q}}, i\nu_{m'}) G(\mathbf{k} + \mathbf{Q} + \tilde{\mathbf{q}}, i\nu_{m'} + i\omega_n) F_{OZ}(\mathbf{Q} + \tilde{\mathbf{q}}, i\nu_{m'} - i\nu_m), \quad (10)$$

where we again focus only on the long-wavelength limit $\mathbf{q} \to 0$.

Analogously as for the bubble contribution, we evaluate in the following $\chi_{VERT}(i\omega_n)$ only for non-negative Matsubara frequencies $\omega_n \geq 0$. This implies again $\omega_n > -\nu_m > 0$, as well as $G(\mathbf{k}, i\nu_m) \to G^h(\mathbf{k}, i\nu_m)$, and $G(\mathbf{k}, i\nu_m + i\omega_n) \to G^e(\mathbf{k}, i\nu_m + i\omega_n)$. To determine the causality properties of the remaining two Green's functions in Eq. (10), we utilize the particle conservation law. With the initial assumption of $\omega_n \geq 0$, the only way to comply with the particle conservation law is to restrict $\nu_{m'}$ to $\omega_n > -\nu_{m'} > 0$, which automatically imposes $G(\mathbf{k}+\mathbf{Q}+\tilde{\mathbf{q}}, i\nu_{m'}) \to G^h(\mathbf{k}+\mathbf{Q}+\tilde{\mathbf{q}}, i\nu_{m'})$, $G(\mathbf{k}+\mathbf{Q}+\tilde{\mathbf{q}}, i\nu_{m'}+i\omega_n) \to G^e(\mathbf{k}+\mathbf{Q}+\tilde{\mathbf{q}}, i\nu_{m'}+i\omega_n)$. Following the visual representation of a dominant $\pi$-ton process depicted in Fig. 2, we note that these latter constraints on the Green's functions are feasible due to the finite scattering rate $\tau^{-1}$ which smears the fermion states around the Fermi surface. This ensures that the scattered hole (electron) is within reach of the hole-like (electron-like) states. Otherwise, the entire contribution would vanish at finite frequencies, similar to how the Drude contribution collapses to a delta function at zero frequency when there are no momentum relaxation processes.

In order to proceed, we note that the $\tilde{\mathbf{q}}$ dependence in Green's functions is weak in comparison with the $\tilde{\mathbf{q}}$ dependence of the vertex function in Eq. (9), so we keep it only in the vertex function peaked at $|\tilde{\mathbf{q}}| \approx 0$, decoupling thus the two momentum summations. Furthermore, for analogous reasons, we set $\nu_{m'} = \nu_m$ in Green's functions while retaining for now the full dependence on $\nu_m$ and $\nu_{m'}$ in the vertex function. For the $\pi$-ton vertex contributions to the current-current correlation function, we thus have for $\xi \gg 1$

$$\chi_{VERT}(i\omega_n) = -\frac{2}{\beta^2} \sum_{\omega_n > -\nu_m, -\nu_{m'} > 0} \frac{1}{N} \sum_{\mathbf{k}} \nu_{\mathbf{k}} \nu_{\mathbf{k}+\mathbf{Q}} G^h(\mathbf{k}, i\nu_m) G^e(\mathbf{k}, i\nu_m + i\omega_n)$$
$$\times G^h(\mathbf{k}+\mathbf{Q}, i\nu_m) G^e(\mathbf{k}+\mathbf{Q}, i\nu_m + i\omega_n) \frac{1}{N} \sum_{\tilde{\mathbf{q}}} F_{OZ}(\mathbf{Q} + \tilde{\mathbf{q}}, i\nu_{m'} - i\nu_m). \quad (11)$$

---

[1] Single-boson vertex corrections have also been considered in other calculations, for example in the context of slave boson theories [50] where, however, the dominating physics is a resonant valence bond (RVB) [51] instead of a divergent antiferromagnetic correlation length. Hence the physics and contribution to the optical conductivity is very different [50].

Lastly, we use the eightfold symmetry depicted in Fig. 2 to set $G^h(\mathbf{k}+\mathbf{Q}, i\nu_m) \to G^h(\mathbf{k}, i\nu_m)$, $G^e(\mathbf{k}+\mathbf{Q}, i\nu_m + i\omega_n) \to G^e(\mathbf{k}, i\nu_m + i\omega_n)$, and importantly $v_{\mathbf{k}+\mathbf{Q}} \to -v_{\mathbf{k}}$ in Eq. (11), leading finally to

$$\chi_{VERT}(i\omega_n) = \frac{2}{\beta^2} \sum_{\omega_n > -\nu_m > 0} \frac{1}{N} \sum_{\mathbf{k}} v_{\mathbf{k}}^2 \left[ G^h(\mathbf{k}, i\nu_m) \right]^2 \left[ G^e(\mathbf{k}, i\nu_m + i\omega_n) \right]^2$$
$$\times \sum_{\omega_n > -\nu_{m'} > 0} \frac{1}{N} \sum_{\tilde{\mathbf{q}}} F_{OZ}(\mathbf{Q} + \tilde{\mathbf{q}}, i\nu_{m'} - i\nu_m). \tag{12}$$

Just as with the bubble contribution, we use the fact that the dominant contributions to the current fluctuations come from the states at the Fermi level, allowing us to again replace the summation over $\mathbf{k}$ with an energy integral. Using further the assumed form of the Green's function in Eq. (1), this yields

$$\frac{1}{N} \sum_{\mathbf{k}} v_{\mathbf{k}}^2 \left[ G^h(\mathbf{k}, i\nu_m) \right]^2 \left[ G^e(\mathbf{k}, i\nu_m + i\omega_n) \right]^2$$
$$\approx g(\varepsilon_F) \langle v^2 \rangle_{FS} \int_{-\infty}^{+\infty} d\varepsilon \left[ \frac{1}{i\nu_m - \varepsilon - \frac{i}{2\tau}} \right]^2 \left[ \frac{1}{i\nu_m + i\omega_n - \varepsilon + \frac{i}{2\tau}} \right]^2. \tag{13}$$

This integral over $\varepsilon$ can be readily evaluated in the complex plane, as outlined in Appendix A, resulting in

$$\int_{-\infty}^{+\infty} d\varepsilon \left[ \frac{1}{i\nu_m - \varepsilon - \frac{i}{2\tau}} \right]^2 \left[ \frac{1}{i\nu_m + i\omega_n - \varepsilon + \frac{i}{2\tau}} \right]^2 = -4\pi i \frac{1}{\left( i\omega_n + \frac{i}{\tau} \right)^3}, \tag{14}$$

which indicates that the $\nu_m$ dependence in the Green's functions has again been lost, appearing only in the vertex function, giving

$$\chi_{VERT}(i\omega_n) = -\frac{8\pi i}{\beta^2} g(\varepsilon_F) \langle v^2 \rangle_{FS} \frac{1}{\left( i\omega_n + \frac{i}{\tau} \right)^3} \sum_{\omega_n > -\nu_m, -\nu_{m'} > 0} \frac{1}{N} \sum_{\tilde{\mathbf{q}}} F_{OZ}(\mathbf{Q} + \tilde{\mathbf{q}}, i\nu_{m'} - i\nu_m). \tag{15}$$

What remains is to evaluate the summations over the Ornstein-Zernike vertex function

$$S_{OZ} = \frac{1}{N} \sum_{\tilde{\mathbf{q}}} \sum_{\omega_n > -\nu_m, -\nu_{m'} > 0} F_{OZ}(\mathbf{Q} + \tilde{\mathbf{q}}, i\nu_{m'} - i\nu_m)$$
$$= \frac{1}{N} \sum_{\tilde{\mathbf{q}}} \sum_{\omega_n > -\nu_m, -\nu_{m'} > 0} \frac{A}{\xi^{-2} + \tilde{q}^2 + \lambda |\nu_{m'} - \nu_m|}. \tag{16}$$

We begin by addressing the Matsubara frequency sums. For this purpose, we separate the diagonal, $m = m'$, and the non-diagonal, $m \neq m'$, contributions, where for the diagonal contribution $S_{OZ}^{m=m'} = \frac{n}{N} \sum_{\tilde{\mathbf{q}}} \frac{A}{\xi^{-2} + \tilde{q}^2}$ trivially follows. Here, $n$ is the number of negative Matsubara frequencies $\nu_m$ included in the sum where $|\nu_m| < \omega_n$.

To tackle the non-diagonal contribution, in Fig. 3(a) we plot the Ornstein-Zernike vertex function summed over the Matsubara frequencies $\nu_m$ and $\nu'_m$ for different numbers $n$ of the largest negative Matsubara frequencies included in the sums for $\tilde{q} = 0$. Specifically, we present the total contribution (dark blue line), the diagonal $m = m'$ (light blue line), and the non-diagonal $m \neq m'$ contribution (light green line) for the inverse temperature $\beta = 22$ and Ornstein-Zernike parameters $\xi = 38$, $A = 0.5$, and $\lambda = 0.21$ [32]. For this set of parameters and the considered values of $n$, we observe that the total sum is predominantly determined by the diagonal contribution. Interestingly, however, we empirically find that in the case $\xi \gg 1$

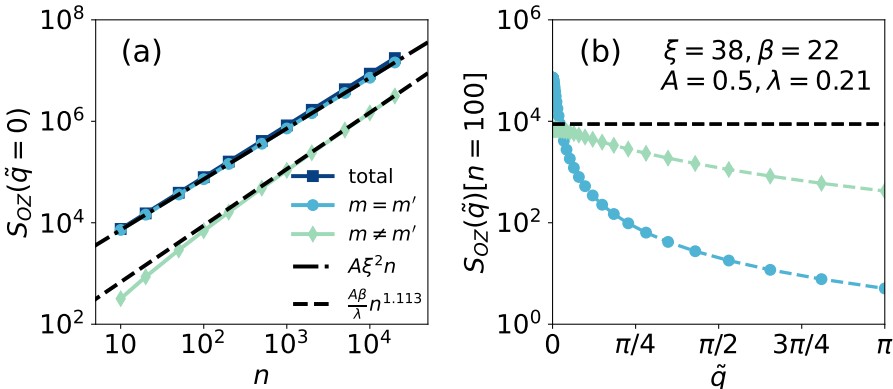

Fig. 3: (a) Total (dark blue line), diagonal $m = m'$ (light blue line), and non-diagonal $m \neq m'$ (light green line) Matsubara frequency summation contributions over the Ornstein-Zernike vertex function, Eq. (16), at $\tilde{q} = 0$ for different numbers of the largest negative Matsubara frequencies $\nu_m$ and $\nu'_m$ taken in the sums. The black dashed-dotted line represents the diagonal contribution equal to $nA\xi^2$, while the black dashed line represents the upper bound on the non-diagonal contribution of $n^c \frac{A\beta}{\lambda}$ with $c \approx 1.113$ for considered $n$ values. (b) Momentum dependence of the diagonal, $m = m'$ (light blue), and non-diagonal, $m \neq m'$ (light green), Matsubara frequency summation contributions over the Ornstein-Zernike vertex function, summed over $n = 100$ largest negative Matsubara frequencies $\nu_m$ and $\nu'_m$. The inverse temperature is $\beta = 22$, while the Ornstein-Zernike parameters read $\xi = 38$, $A = 0.5$, and $\lambda = 0.21$ [32].

the non-diagonal contribution is bounded by $n^c \frac{A\beta}{\lambda}$ with $c \approx 1.113$, see Fig. 3(b), for the considered $n$ values. This bound is approached for $\tilde{q} = 0$ already when $n \sim 10^3$ as indicated in Fig. 3(a). The latter observation suggests that as $n \to \infty$, the non-diagonal contribution may eventually surpass the diagonal term. Nevertheless, it is important to recall that the Matsubara frequency sums are limited by $\omega_n > -\nu_m, -\nu_m > 0$, which restricts $n$ to finite values. Considering the condition $\xi \gg 1$, which holds close to the paramagnetic-to-antiferromagnetic transition boundary, we can then roughly estimate the relative importance of the diagonal and non-diagonal contributions by comparing $S_{OZ}^{m=m'}/n = A\xi^2$ and $S_{OZ}^{m\neq m'}/n^a \approx A\beta/\lambda$. In particular, providing that $\xi \gg \sqrt{\frac{\beta}{\lambda}}$ roughly holds, we can approximate the total Matsubara frequency sums by considering only the diagonal contribution. Assuming the weak temperature dependence of $\lambda$ [31,32], this translates to the correlation length growing faster than $\beta^{\frac{1}{2}} = T^{-\frac{1}{2}}$. In all the subsequent cases this condition will indeed be met, so we keep only the diagonal contribution in Eq. (16), i.e., $S_{OZ} \approx S_{OZ}^{m=m'}$.

Let us finish the discussion about the Matsubara frequency sums by recalling again that they are restricted by $\omega_n > -\nu_m, -\nu'_m > 0$. This implies that in the evaluation of $\chi_{VERT}(i\omega_n)$ for a given $\omega_n$, $n$ in $S_{OZ}^{m=m'}$ is the index of the corresponding bosonic Matsubara frequency, $i\omega_n = i\frac{2\pi n}{\beta}$, analogously as was the case for the bubble contribution. Taking that into account, we then have

$$\chi_{VERT}(i\omega_n) \approx -\frac{4g(\varepsilon_F)\langle v^2 \rangle_{FS}}{\beta} \frac{i\frac{2\pi n}{\beta}}{\left(i\omega_n + \frac{i}{\tau}\right)^3} \frac{S_{OZ}^{m=m'}}{n} = -\frac{4g(\varepsilon_F)\langle v^2 \rangle_{FS}}{\beta} \frac{i\omega_n}{\left(i\omega_n + \frac{i}{\tau}\right)^3} s_{OZ}^{m=m'}, \quad (17)$$

where we have introduced $s_{OZ}^{m=m'} \equiv \frac{S_{OZ}^{m=m'}}{n} = \frac{1}{N}\sum_{\tilde{\mathbf{q}}} \frac{A}{\xi^{-2}+\tilde{q}^2}$. Analytic continuation, $i\omega_n \to \omega + i\eta$,

now gives

$$\chi_{VERT}(\omega) = -\frac{4g(\varepsilon_F)\langle v^2 \rangle_{FS}}{\beta} \frac{\omega}{\left(\omega + \frac{i}{\tau}\right)^3} s_{OZ}^{m=m'}, \tag{18}$$

and correspondingly

$$\mathrm{Im}\chi_{VERT}(\omega) = 2\frac{2g(\varepsilon_F)\langle v^2 \rangle_{FS}\, \tau}{\beta} \tau^2 \frac{\left(3\omega^2\tau^2 - 1\right)\omega}{(1+\omega^2\tau^2)^3} s_{OZ}^{m=m'}, \tag{19}$$

so we get for the optical conductivity $\sigma(\omega) = \frac{\mathrm{Im}\chi(\omega)}{\omega}$

$$\sigma_{VERT}(\omega) = -2\frac{\sigma_{BUB}^{dc}}{\beta} \tau^2 \frac{1-3\omega^2\tau^2}{(1+\omega^2\tau^2)^3} s_{OZ}^{m=m'}. \tag{20}$$

In Appendix B, we evaluate the summation over the momentum $\tilde{\mathbf{q}}$ in $s_{OZ}^{m=m'}$ for the 1D and 2D cases; if we use the same modeling for 3D, the corrections will be very small due to momentum integration. This yields the $\pi$-ton vertex contributions to the optical conductivity

$$\sigma_{VERT}(\omega) = -A\,T\tau^2\sigma_{BUB}^{dc}\frac{1-3\omega^2\tau^2}{(1+\omega^2\tau^2)^3} \begin{cases} \xi/\pi, & \text{in 1D,} \\ \ln(\pi\xi)/\pi^3, & \text{in 2D,} \end{cases} \tag{21}$$

which is the main result of our paper.

## 3 Discussion

To help us keep track of upcoming discussions, it is convenient to introduce

$$\sigma_{VERT}^{dc} \equiv -A\,T\tau^2\sigma_{BUB}^{dc} \begin{cases} \xi/\pi, & \text{in 1D,} \\ \ln(\pi\xi)/\pi^3, & \text{in 2D} \end{cases} < 0, \tag{22}$$

keeping in mind that $\xi \gg 1$. This $\sigma_{VERT}^{dc}$ gives the dc value of the $\pi$-ton vertex contributions, which is negative and thus suppresses the Drude conductivity at small frequencies. Please note that the result for the 2D case also suggests an enhancement of the dc conductivity far from the transition when $\xi \ll 1$, similar to the high-temperature findings in Refs. [31, 52]. However, this regime is not the focus of our analysis in this work. For large frequencies, on the other hand, $3\omega^2\tau^2$ in the numerator of Eq. (21) becomes large, and correspondingly $\sigma_{VERT}(\omega)$ is positive and asymptotically decays to zero as

$$\lim_{\omega \to \infty} \sigma_{VERT}(\omega) = \frac{3}{\omega^4\tau^4}\left|\sigma_{VERT}^{dc}\right|. \tag{23}$$

This further implies that the sign of the $\pi$-ton vertex contributions changes at the zero of Eq. (21), $\sigma_{VERT}(\omega_0) = 0$, which is given solely by the fermion lifetime, $\omega_0 = \frac{1}{\sqrt{3}}\tau^{-1}$. Additionally, the maximum of $\sigma_{VERT}(\omega)$ is at the frequency $\omega_{MAX} = \tau^{-1}$, with the value of the maximum $\sigma_{VERT}(\omega_{MAX}) = \frac{1}{4}\left|\sigma_{VERT}^{dc}\right|$ independent of dimension.

Such shape of the $\pi$-ton vertex contributions together with the Drude contribution may result in the DDP in the total optical conductivity, $\sigma_{TOT}(\omega) = \sigma_{BUB}(\omega) + \sigma_{VERT}(\omega)$. Given that we have closed-form analytical expressions for both contributions, we can determine the criterion for the DDP appearance, as well as the DDP frequency and height. The details of these

calculations can be found in Appendix C, while here we just highlight the final expression for the DDP frequency

$$\omega_{DDP} = \frac{1}{\tau} \sqrt{\sqrt{3 \frac{|\sigma_{VERT}^{dc}|}{\sigma_{BUB}^{dc}} \left(3 \frac{|\sigma_{VERT}^{dc}|}{\sigma_{BUB}^{dc}} + 4\right)} - \left(1 + 3 \frac{|\sigma_{VERT}^{dc}|}{\sigma_{BUB}^{dc}}\right)}. \qquad (24)$$

The Drude peak will be displaced to the finite frequency $\omega_{DDP}$ when there exists a real solution of Eq. (24), which is given by the criterion $6|\sigma_{VERT}^{dc}| > \sigma_{BUB}^{dc}$. In the case when the $\pi$-ton contributions become particularly strong, $6|\sigma_{VERT}^{dc}| \gg \sigma_{BUB}^{dc}$, Eq. (24) indicates that the DDP frequency would be determined solely by the fermion lifetime $\omega_{DDP} \approx \tau^{-1}$, giving for the DDP height $\sigma_{TOT}(\omega_{DDP}) = \sigma_{BUB}^{dc} - \frac{1}{4}|\sigma_{VERT}^{dc}|$. Note, however, that $|\sigma_{VERT}^{dc}| > \sigma_{BUB}^{dc}$ can give an unphysical negative dc conductivity, so the criterion for the applicability of our results and the appearance of the DDP can be put into the inequality $\sigma_{BUB}^{dc}/6 < |\sigma_{VERT}^{dc}| < \sigma_{BUB}^{dc}$. In reality, if the vertex corrections in the particle-hole channel become that large, self-consistency effects will become large and relevant, too, namely (i) a dampening of the self-energy and (ii) the parquet coupling between different channels. These effects are not included in our calculations, and eventually need to yield a positive conductivity.

Finally, we would especially like to emphasize that the $\pi$-ton vertex contributions in Eq. (21) comply with the optical sum rule following from the Ward identities [53,54]. Namely, with a Green's function like the one in Eq. (1) featuring the momentum-independent self-energy $\Sigma = -\frac{i}{2\tau} \text{sgn} \nu_m$, the full optical spectral weight is entirely given by the bubble contribution. By integrating the $\pi$-ton vertex contributions in Eq. (21) over frequencies it readily follows that the corresponding optical spectral weight vanishes, i.e., $\int_0^\infty d\omega\, \sigma_{VERT}(\omega) = 0$. Thus, such contributions may only shift, but not add any additional optical spectral weight.

## 3.1 Comparison of analytical and adaptive integration results

Before going into the consideration of DDP caused by $\pi$-tons in the 2D case, we first benchmark our analytical results, Eqs. (21) and (22), against those obtained from the adaptive integration of the $\pi$-ton vertex contributions formulated on the real frequency axis. In Refs. [31,32], Eq. (8) was analytically continued to the real frequency axis for a simplified vertex function $F_d$, that depends only on one frequency $i\nu_{m'} - i\nu_m$ and momentum $\mathbf{k}' - \mathbf{k}$ (the complete expression for the current-current correlation function on the real frequency axis is given in Ref. [31] and in Appendix A of Ref. [32]). The two Matsubara sums of Eq. (8) become integrals that can be handled, together with the momentum integrals, by adaptive integration once analytical expressions for the Green's functions and vertex are known. In particular, we assume here the Ornstein-Zernike form of the vertex function , whose parameters are obtained within the RPA, as described in detail in Ref. [32]. The Green's functions are given by the analytic continuation of Eq. (1). We compare separately the dc values, $\sigma_{VERT}^{dc}$, as well as the full frequency dependence of the $\pi$-ton vertex contributions. The two quantities are shown in Figs. 4(a,b) and (c,d) for the 1D case and the 2D case, respectively.

### 3.1.1 1D case

For the 1D case, the parameterization of the Ornstein-Zernike vertex function within the RPA with the Hubbard interaction $U = 2$ and a quite generic Fermi liquid-like scattering rate $\tau^{-1} = 0.1547 + 1.637\, T^2$ as well as the adaptive integration of the corresponding $\pi$-ton vertex contributions has been already carried out in Ref. [32]. In such an approximation, the correlation length increases algebraically with $T$ approaching $T_c$: $\xi \sim (T - T_c)^{-\nu}$, where $T_c \approx 1/23$ for these specific parameters. Please note that we assume finite $\tau$ at $T_c$ throughout the paper. These numerical results for the frequency dependence of the $\pi$-ton vertex contributions

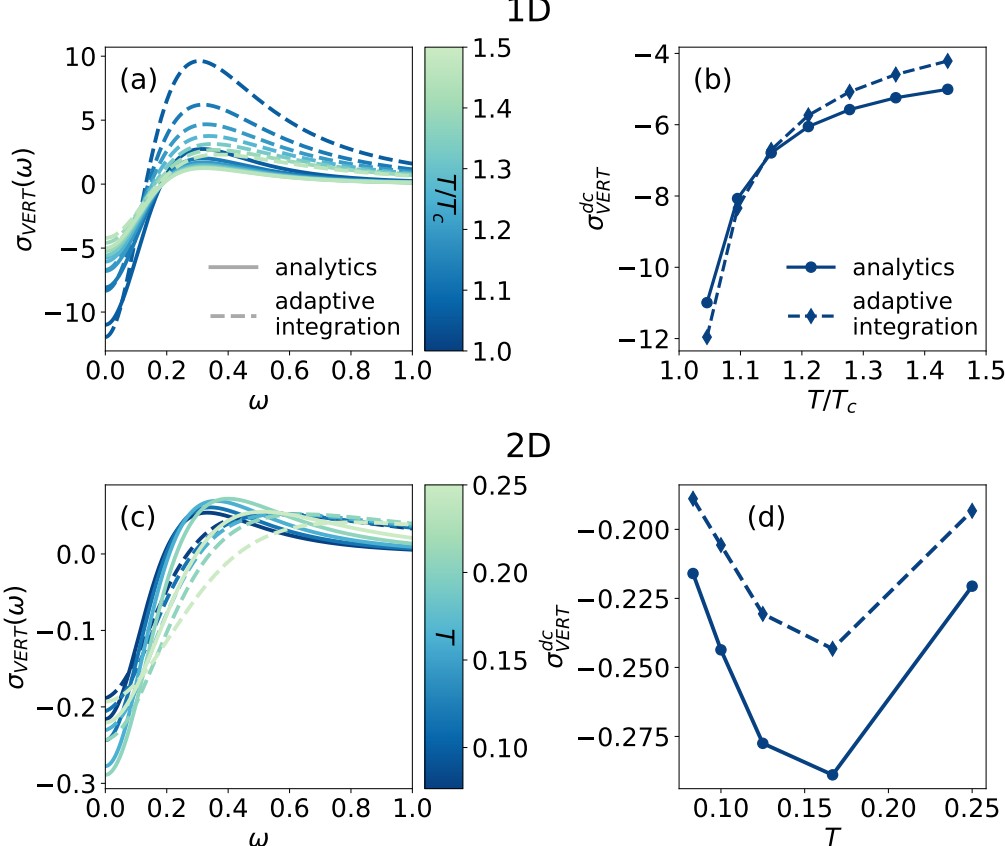

Fig. 4: (a,c) Analytical results [solid lines, Eq. (21)] vs adaptive integration [dashed lines] for the vertex correction to the optical conductivity. Several temperatures are discriminated by color. (b,d) dc value of the vertex correction as a function of temperature [Eq. (22)]. Here, the temperature dependencies of the fermion lifetimes and Ornstein-Zernike parameters have been taken from Ref. [32] in the 1D case (a,b) and from Ref. [31] in the 2D case (c,d).

and their dc values are shown as dashed lines for temperatures approaching the transition temperature $T_c$ in Figs. 4(a) and (b), respectively. For the same set of parameters, the analytical results in Eqs. (21) and (22) are shown for comparison as solid lines in Fig. 4(a) and in Fig. 4(b), respectively.

By focusing first on the dc values in Fig. 4(b), we note that our analytical results show excellent qualitative and quantitative agreement with the numerical results obtained by adaptive integration. The two lie almost on top of each other near the transition temperature. As the frequency increases, both results predict a sign change of $\pi$-ton vertex contributions and a broad maximum at some finite frequency, see Fig. 4(a). However, both the position and height of the maximum differ between the two calculations, where the analytical calculations show the maximum at a lower frequency with a significantly smaller value. Such discrepancies at larger frequencies are not so surprising since in the analytical evaluation we considered only small energy transfer processes, giving a better description of $\pi$-ton vertex contributions around the dc values. Overall, taking into account the number of approximations and simplifications imposed in evaluating the analytical results, the qualitative frequency behavior of the $\pi$-ton vertex contributions and their dc values match astonishingly well with the adaptive integration results in the 1D case.

### 3.1.2  2D case

Next, we compare analytical and adaptive integration results for the 2D case. As in the 1D case, we take for the fermion lifetime $\tau^{-1} = 0.1547 + 1.637\,T^2$ disregarding possible differences between the scattering rate in 1D and 2D. At this point, we should note that this temperature dependence of $\tau$ stems from the fitting of the quasiparticle peak to the 2D parquet DΓA results [29,31]. For this specific temperature dependence of $\tau$, the 2D Ornstein-Zernike parameters within the RPA and for $U = 1.9$ have been fitted as thoroughly discussed in Ref. [31]. Here, we only briefly outline the end result (see also Appendix D)

$$\xi = \frac{0.30}{T} + 10^{-3}\exp\frac{0.51}{T}\,, \quad A = 0.41 + 13T^{1.03}\,, \quad \text{and} \quad \lambda = 0.38 + 10.6T^{1.29}\,. \tag{25}$$

It should be noted that the correlation length $\xi$ in Eq. (25), although derived using the RPA with nominally algebraic temperature dependence of the correlation length, was fitted in Ref. [31] with an exponential function. This fitting resembles the ideal 2D zero temperature phase transition behavior according to the Mermin-Wagner theorem [55], with the exponential divergence of the correlation length as zero temperature is approached. The scenario of a finite $T_c$ with the algebraic temperature dependence of the correlation length, as is the case in our 1D modeling, is for 2D also discussed in Sec. 3.2.2.

Integrating the $\pi$-ton vertex contributions over four momenta and two frequencies in the real frequency formulation [31,32] poses significant challenges in ensuring proper convergence in the 2D case, even when using adaptive integration methods. Here, we overcome these challenges using the vegas method from the cuba package [41]. The results of this adaptive integration for the frequency dependence of the $\pi$-ton vertex contributions for several temperatures are shown in Fig. 4(c) with dashed lines, while their dc values with the blue line in Fig. 4(d). For the same 2D parameter set ($\tau$, $\xi$, $A$, and $\lambda$), the corresponding analytical results are shown as full lines in Fig. 4(c) and as solid green lines Fig. 4(d).

For the dc values, there is again an excellent qualitative agreement between the analytical and adaptive integration results regarding the temperature dependence. Quantitatively, though, the analytical values are slightly larger. One possible explanation for this lies in the velocity factor, see Appendix B, which favors momenta in the nodal region of the Fermi surface, while the analytical evaluation assumed a constant value across the whole Fermi surface equal to the average fermion velocity. Similarly like in the 1D case, the discrepancies are also present in the high-frequency regions, where now analytical results predict slightly larger values of the $\pi$-ton vertex contributions maxima. It is essential to highlight, however, that the overall magnitude of the $\pi$-ton vertex contributions agree well in both analytical and adaptive integration calculations. This agreement is particularly important when comparing the 1D and 2D cases, where the analytical calculations corroborate the orders of magnitude differences in $\pi$-ton contributions between the two cases, as previously reported numerically in Ref. [32]. This then carries important implications for the potential formation of the DDP in the 2D case, as discussed further in Sec. 3.2 below.

### 3.2  2D $\pi$-ton vertex contributions

Because the 2D $\pi$-ton vertex contributions shown in Fig. 4(c) are of relatively small magnitude, their inclusion to the Drude optical conductivity results only in a broadening of the Drude peak. This is depicted in Figs. 5(a)-(c), showing the Drude contribution of the bubble term, the $\pi$-ton vertex contributions, and their sum for several temperatures, respectively. Unlike the 1D case, where the $\pi$-ton contributions continuously increase in magnitude as the temperature decreases [32], the 2D $\pi$-ton vertex contributions in Figs. 5(b) initially increase in magnitude with decreasing temperature, though not enough to produce the DDP, but eventually, these

contributions begin to get weaker and finally saturate as zero temperature is approached. This is due to the distinctive characteristics of the 2D case, where the $\pi$-ton contributions depend logarithmically on the correlation length [$\ln(\pi\xi)$ in Eq. (21)]. The correlation length $\xi$ in turn exponentially diverges as the temperature approaches zero, $\xi \sim \exp(1/T)$ [Eq. (25)]. Combining both gives a factor $\beta$, which however cancels out the factor $\beta$ in the denominator of Eq. (21) coming from the Matsubara sums. At the same time, Eq. (25) suggests that $A$ and $\lambda$ go to constant as $T \to 0$, leaving the 2D $\pi$-ton vertex contributions temperature independent at low temperatures, thus explaining the saturation of the $\pi$-ton contributions.

The 2D (temperature behaviors of) Ornstein-Zernike parameters in Eq. (25) apparently proved not to yield large enough $\pi$-ton contributions to result in the DDP. However, since we have a closed analytical expression for $\pi$-tons, we can tweak the Ornstein-Zernike parameters in such a way as to give larger $\pi$-ton contributions and explore possible routes for the appearance of the DDP in the 2D case. Thus, we can identify scenarios where $\pi$-tons are present in 2D systems.

### 3.2.1 Scenario I: Enhanced coupling strength to antiferromagnetic fluctuations

Since the magnitude of the $\pi$-ton contributions is directly proportional to the Ornstein-Zernike parameter $A$, the most straightforward way to increase it is by increasing the parameter $A$. Physically, this corresponds to enhancing the coupling strength $g$ between fermions and AFM fluctuations. In particular, we consider the scenario where the coupling $g$ is uniformly increased by a factor of $\sqrt{2}$ across all temperatures. This results in $A$ being enhanced by a factor of 2, which doubles the magnitude of the $\pi$-ton vertex contributions while maintaining their qualitative temperature behavior as described in Sec. 3.2.

Such enhanced $\pi$-ton vertex contributions are shown in Fig. 5(d) and the resulting total optical conductivity in Fig. 5(e). The Drude peaks in the total optical conductivity are now shifted to finite frequencies, with the DDP frequency being an increasing function of temperature, $\omega_{DDP} \propto T^{\alpha}$, where $0 < \alpha < 1$, as can be seen in Fig. 5(f). Interestingly, as the temperature decreases, the height of the DDP increases, but the overall shape of the DDP becomes less distinct. This behavior is reminiscent of the experimental observations reported in Refs. [1–23], and contrasts with the DDP features associated with $\pi$-tons in the 1D case [32]. In the latter case, as the transition temperature $T_c$ is approached, the displacement of the Drude peak becomes increasingly pronounced. As explained at the beginning of Sec. 3.2, these discrepancies arise due to the peculiar temperature behavior of $\pi$-tons in the 2D case: At low temperatures, the $\pi$-ton contributions reach a saturation point, while the Drude peak of the bubble contribution sharpens, indicating a crossover from the displaced to the broadened Drude peak behavior in the optical conductivity as the temperature decreases in the enhanced coupling strength scenario.

### 3.2.2 Scenario II: Finite temperature phase transition

In Ref. [32], it was reported that the magnitude of the 2D $\pi$-ton vertex contributions increases monotonically and logarithmically as the transition temperature is approached, which is in sharp contrast to the behavior of the 2D $\pi$-ton contributions discussed so far in Secs. 3.2 and 3.2.1. To this end, we note, however, that in Ref. [32], (1) the phase transition appeared at the finite temperature $T_c \approx 1/19$, and (2) the correlation length diverged with a power-law dependence on temperature rather than exponentially as in Eq. (25). To mimic such a scenario, in the following, we modify the temperature behavior of the correlation length in Eq. (25) to $\tilde{\xi} = \frac{0.30}{T - T_c} + 10^{-3} \exp^{\frac{0.51}{T}}$ [we turn back to the original coupling strength with the Ornstein-Zernike parameter $A$ given in Eq. (25)]. Now the power-law term in the correlation length outweighs the exponential term in the vicinity of $T_c$, causing the $\pi$-ton vertex contributions

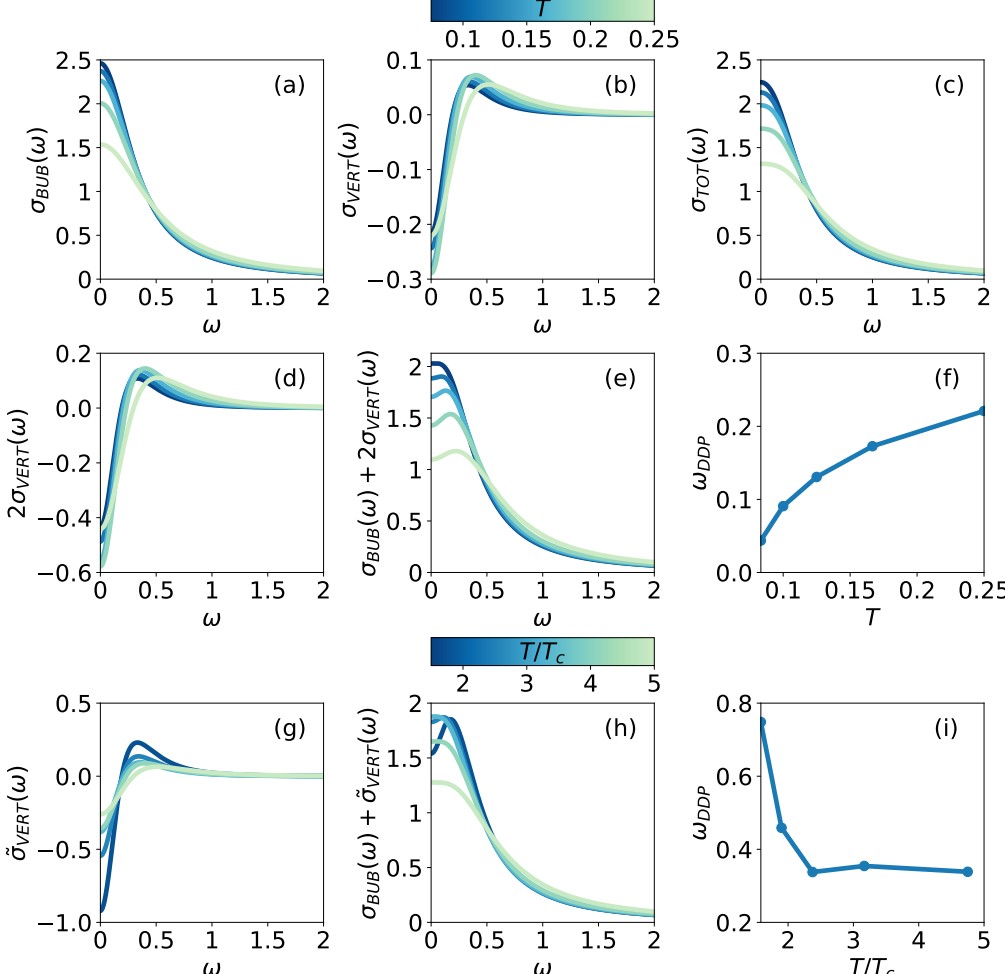

Fig. 5: (a) Bubble, (b) $\pi$-ton vertex, and (c) total contribution to the optical conductivity for several temperatures $T$ for the 2D case with the fermion lifetime $\tau^{-1} = 0.1547 + 1.637\ T^2$ and the Ornstein-Zernike parameters in Eq. (25). (d, e) $\pi$-ton contribution and total optical conductivity for Scenario I: twice the magnitude $A$ of the $\pi$-ton vertex contributions. (g, h) $\pi$-ton contribution and total optical conductivity for Scenario II: a finite temperature phase transition at $T_c \approx 1/19$ and a power-law divergence of the correlation length, $\tilde{\xi} \sim (T - T_c)^{-1}$. (f, i) Temperature dependence of the displaced Drude peak frequency for the latter two cases.

to diverge logarithmically with temperature according to Eq. (21) as $\sigma_{VERT} \propto \ln(T - T_c)$. Analogously then, as in Ref. [32], the magnitude of $\pi$-ton vertex contributions monotonically increases all the way down to the transition boundary as shown in Fig. 5(g).

Similar to the enhanced coupling strength scenario, the corresponding total optical conductivity in Fig. 5(h) also features the DDP, but with a qualitatively different temperature behavior compared to that in Fig. 5(e) and discussed in Sec. 3.2.1. Specifically, at high temperatures, the $\pi$-ton vertex contributions are small, leading only to the broadening of the Drude peak. As the temperature decreases, the $\pi$-ton contributions eventually become strong enough to displace the Drude peak, which becomes more pronounced as the temperature approaches $T_c$. Notably, Fig. 5(i) illustrates that in this scenario, starting from low temperatures, the DDP frequency initially decreases as the temperature increases, then increases within an intermediate temperature range, before decreasing again at higher temperatures.

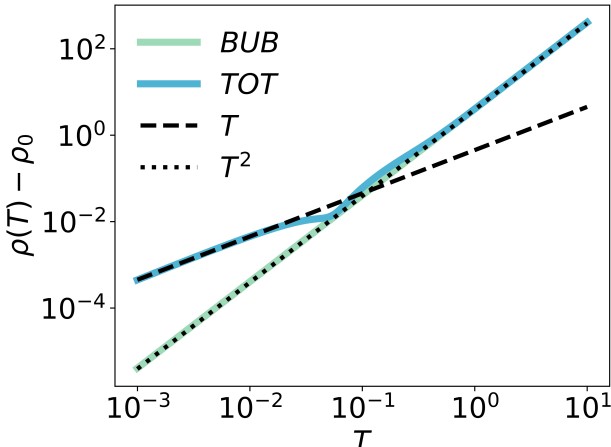

Fig. 6: Temperature dependence of the dc resistivity calculated from the bubble contribution (green line) and the total contribution (bubble + $\pi$-ton vertex) (blue line), with the residual resistivity subtracted in both cases. Including the $\pi$-ton contributions, the dc resistivity exhibits a crossover from the linear $T$ behavior at low temperatures to Fermi liquid-like $T^2$ dependence at high temperatures. For reference, we display the $T$ (black dashed line) and $T^2$ (black dotted line) lines to guide these temperature behaviors.

Finally, we want to emphasize that while the overall qualitative frequency behavior of the $\pi$-ton vertex contributions does not depend on the peculiarities of the Ornstein-Zernike parameters, their qualitative and quantitative temperature dependence does. These temperature dependencies are primarily governed by the correlation length, which allows for various scenarios for the appearance of the DDP due to the $\pi$-ton contributions in the 2D case. Here we consider just two of them. Scenario I with a relatively strong coupling of fermions to AFM fluctuations and an exponential divergence of the correlation length at zero temperature. In this scenario, the displacement of the Drude peak appears at high temperatures and gradually transitions to a simple broadening of the Drude peak at low temperatures with the DDP frequency increasing with temperature as in some experiments [1–23]. In Scenario II, with a finite temperature phase transition and a power-law divergence of the correlation length at a finite $T_c$, the magnitude of the $\pi$-ton vertex contributions continuously increases as $T_c$ is approached. A DDP is observed in such a case, however, only close to $T_c$.

### 3.2.3 Temperature dependence of the dc resistivity

The temperature dependence of the dc resistivity can be significantly affected by the temperature dependence of the Ornstein-Zernike parameters. To illustrate this, we consider an ideal 2D system exhibiting a zero-temperature phase transition, where the correlation length diverges exponentially as $\xi \approx p \exp^{r/T}$. Furthermore, following Eq. (25), we assume $A \approx c + dT$. Here, $c$, $d$, $p$, and $r$ are temperature independent constants.

Focusing solely on the bubble contribution for a moment, the DC resistivity is straightforwardly given by $\rho_{BUB}^{dc} = \left[\sigma_{BUB}^{dc}\right]^{-1} = \left[2g(\varepsilon_F)\left\langle v^2\right\rangle_{FS}\tau\right]^{-1}$. Assuming that $g(\varepsilon_F)$ and $\left\langle v^2\right\rangle_{FS}$ have weak temperature dependencies, the temperature dependence of the resistivity is then solely governed by the scattering rate $\rho_{BUB}^{dc} \sim \tau^{-1}$. With the Fermi liquid-like assumption of $\tau^{-1} \sim a + bT^2$ this then leads to the Fermi liquid-like $T^2$ resistivity. This is illustrated in Fig. 6 with the green line, depicting the temperature dependence of the bubble contribution to the dc resistivity after subtracting the constant residual resistivity $\rho_0$.

The situation with the $\pi$-ton vertex contribution is a bit more involved. With the above assumption for the temperature behavior of the Ornstein-Zernike $A$ and $\xi$ in the ideal 2D case, Eq. (22) together with the bubble contribution gives

$$\rho_{TOT} = \sigma_{TOT}^{-1} = \left[\sigma_{BUB}^{dc} + \sigma_{VERT}^{dc}\right]^{-1} = \frac{1}{\sigma_{BUB}^{dc}\left(1 - A\,T\,\tau^2\ln(\pi\xi)/\pi^3\right)}$$
$$\sim \frac{a + bT^2}{1 - (c + dT)(a + bT^2)^{-2}(r + T\ln(p\pi))/\pi^3}\,. \tag{26}$$

In the high-temperature limit, $T \to \infty$, the denominator in Eq. (26) approaches 1. As a result, the total resistivity exhibits the Fermi liquid-like $T^2$ dependence, similar to the bubble DC resistivity, due to the assumed Fermi liquid-like behavior of the scattering rate. More generally, we have $\rho_{TOT}(T \to \infty) \sim \tau^{-1}$. Interestingly, however, at low temperatures, $T \to 0$, the resistivity becomes linear-in-$T$, despite the scattering rate exhibiting Fermi liquid-like behavior. This is most easily seen from the above equation by noting the linear-in-$T$ terms in the denominator, $\rho_{TOT}(T \to 0) \to \frac{a + bT^2}{C - DT}$. Here, $C = 1 - \frac{cr}{a^2\pi^3}$ and $D = \frac{dr + c\ln p\pi}{a^2\pi^3}$ are temperature-independent constants, coming from $\tau$, $A$ and $\xi$. To ensure a physical solution, $C - DT > 0$ needs to hold for each $T$. The Taylor expansion for $T \to 0$ now gives $\rho_{TOT}(T \to 0) \approx \frac{a}{C} + \frac{aD}{C^2}T + \frac{b}{C}T^2 + \ldots$, suggesting that the linear-in-$T$ term becomes leading at low enough temperatures, together with the renormalization of the residual resistivity $a \to \frac{a}{C}$. This is demonstrated in Fig. 6, showing the temperature dependence of the total dc resistivity (the residual part $\rho_0$ is again subtracted) for Ornstein-Zernike parameters described by Eq. (25) (the exponential term dominates over the algebraic term at low temperatures). The total dc resistivity clearly demonstrates a crossover from linear $T$ behavior at low temperatures to Fermi liquid-like $T^2$ behavior at high temperatures. Note, however, that this behavior arises from the specific temperature dependencies of the Ornstein-Zernike parameters - different temperature dependencies of $A$ and $\xi$ could lead to alternative temperature dependencies for the resistivity.

## 3.3 $\pi$-ton vs localization vertex contributions

Besides the displacement of the Drude peak through $\pi$-ton vertex contributions, it is well known that the DDP can also arise from the localization vertex corrections [24, 26, 27, 42, 56, 57]. It is therefore fitting to wrap up with a brief discussion of the distinctions between these two types of vertex contributions.

The first key difference arises from the underlying microscopic source giving rise to the vertex corrections. Specifically, the $\pi$-ton vertex contributions stem from interactions of fermions with a soft critical boson (cf. below), whereas the core source of localization vertex corrections is static disorder. It should be, however, emphasized that in the recent studies on transient localization [24, 26, 58], the initial model Hamiltonian did not include disorder from the outset; instead, an effectively disordered environment was generated through fermion-boson interactions. The latter involves the destructive interference of the electron wave function, which, when classified according to the two-particle reducibility, falls into the category of particle-particle reducible vertex contributions. In contrast, the $\pi$-ton vertex contributions are reducible in the transversal particle-hole channel.

This brings us to the second important difference: the topologies of the Feynman diagrams associated with these two types of vertex corrections are distinct. This distinction further affects how momentum and frequency summations couple the fermion Green's functions and the vertex function in the current-current correlation function. In particular, for the $\pi$-ton vertex contributions, the frequency behavior is determined primarily by the fermion Green's functions, as follows from Eqs. (12) and (14), whereas in the case of localization, the vertex function governs the frequency behavior of the optical conductivity [42, 56].

Regarding the vertex function, the third important point is that in both cases, the vertex function resembles the form of an overdamped boson mode. In the context of localization physics, this boson is the diffuson with the form of the vertex function $F(\mathbf{q}, i\omega_m) \sim \frac{1}{D|\mathbf{q}|^2 + |\omega_m|}$ [59], whereas, in the present consideration of $\pi$-ton effects, the boson represents AFM fluctuations with $F(\mathbf{q}, i\omega_m) \sim \frac{1}{\xi^{-2} + |\mathbf{q} - \mathbf{Q}|^2 + \lambda|\omega_m|}$. The crucial difference between these two vertex functions is that the $\pi$-ton vertex function possesses a mass term determined by the correlation length, which is absent in the case of the diffuson. Because of that mass term, the strength of the $\pi$-ton vertex contributions scales with the correlation length of the critical fluctuations, with the specific scaling behavior depending on the system dimension. In the 1D case, the strength of the $\pi$-ton vertex contributions scales linearly, while in the 2D case, it scales logarithmically with the correlation length. This correlation length scaling, along with the relation between Drude peak displacement and proximity to the phase transition, may be used in experiments to discriminate $\pi$-tons from other mechanisms, particularly localization corrections, leading to the DDP.

## 4 Conclusion

A displaced Drude peak originating from $\pi$-tons was recently found numerically in the weakly correlated metallic regime in one dimension near the paramagnetic-to-antiferromagnetic transition boundary [32]. Although qualitatively similar $\pi$-ton contributions have been observed in two dimensions, their logarithmic temperature scaling prevented unambiguous numerical statements.

Here, we derive the analytical expression Eq. (21) for $\pi$-ton vertex contributions and Eq. (24) for the peak position of the DDP, and validate these against an improved, adaptive numerical integration. Our assumptions to arrive at these analytical expressions are that: we are at small frequencies, the correlation length is large, and—in 2D—the optical conductivity mainly stems from nodal momenta where the velocity is largest. We find that a displaced Drude peak due to the $\pi$-tons may appear in 2D systems with a relatively strong coupling of the electrons to antiferromagnetic spin fluctuations or for a finite temperature phase transition. In the former case, the displaced Drude peak gradually diminishes as the temperature decreases, while in the second scenario, the effect will be the opposite.

With the identified characteristic dependencies of the $\pi$-ton vertex contributions in 1D and 2D [see Eq. (21), Eq. (24), and the Abstract], we have laid the foundations for observing $\pi$-tons also in experiments. In 3D, we expect them to be small. We find, as in some experiments, an enhancement of the DDP with increasing temperature if we have the ideal 2D case with an exponential scaling of the correlation length with $1/T$ and strong coupling to spin or charge fluctuations. In this case, we also argue that $\pi$-tons cause the dc resistivity to exhibit a linear $T$ behavior at low temperatures. An even more clear-cut proof would be to study the DDP upon approaching a finite temperature phase transition at $T_c$. In such scenarios, we predict an algebraic, $\sigma_{VERT}^{dc} \propto \xi \sim (T - T_c)^{-\nu}$, and logarithmic, $\sigma_{VERT}^{dc} \propto \ln \xi \sim \nu \ln(T - T_c)$, enhancement of $\pi$-tons in 1D and 2D, respectively, which should be contrasted with $\ln T$ dependence of localization corrections in 2D.

## Acknowledgments

We thank M. Grilli, O. Simard, P. Werner, and P. Worm for useful discussions.

**Funding information** We acknowledge the support of the Austrian Science Fund (FWF) Grant DOI 10.55776/P36213 and 10.55776/V1018.

**Data availability** The raw data presented in the manuscript is available at https://doi.org/10.48436/8tzsb-xpk78.

## A Contour integrations

We solve the integral

$$\mathcal{I}_1 = \int_{-\infty}^{+\infty} d\varepsilon \left[ \frac{1}{i\omega_n + iv_m - \varepsilon + \frac{i}{2\tau}} \right] \left[ \frac{1}{iv_m - \varepsilon - \frac{i}{2\tau}} \right] = \int_{-\infty}^{+\infty} d\varepsilon \, u_1(\varepsilon), \qquad (A.1)$$

appearing in the bubble contribution to the current-current correlation function, Eq. (5), by performing the contour integration in the complex plane. In particular, we choose the contour $\mathcal{C}$ running from $+\infty$ to $-\infty$ on the real axis enclosed by the arc $\Gamma$ of infinite radius in the lower half of the complex plane, so that we can write

$$\oint_{\mathcal{C}} dz \, u_1(z) = -\mathcal{I}_1 + \int_{\Gamma} dz \, u_1(z) = 2\pi i \operatorname{Res}\left( u_1, iv_m - \frac{i}{2\tau} \right). \qquad (A.2)$$

Note that $z = iv_m - \frac{i}{2\tau}$ is the only pole (of order one) in the lower half of the complex plane due to the constraint $\omega_n > -v_m > 0$. The arc that closes the contour does not give a contribution because the integrand is decaying faster than $1/|z|$, while the corresponding residue reads

$$\operatorname{Res}\left( u_1, iv_m - \frac{i}{2\tau} \right) = \lim_{z \to iv_m - \frac{i}{2\tau}} \left[ z - \left( iv_m - \frac{i}{2\tau} \right) \right] \left[ \frac{1}{i\omega_n + iv_m - z + \frac{i}{2\tau}} \right] \left[ \frac{1}{iv_m - z - \frac{i}{2\tau}} \right]$$

$$= -\frac{1}{i\omega_n + \frac{i}{\tau}}. \qquad (A.3)$$

This yields the required integral

$$\mathcal{I}_1 = \frac{2\pi i}{i\omega_n + \frac{i}{\tau}}. \qquad (A.4)$$

For the evaluation of the $\pi$-ton vertex contributions, we additionally need to compute

$$\mathcal{I}_2 = \int_{-\infty}^{+\infty} d\varepsilon \left[ \frac{1}{iv_m + i\omega_n - \varepsilon + \frac{i}{2\tau}} \right]^2 \left[ \frac{1}{iv_m - \varepsilon - \frac{i}{2\tau}} \right]^2 = \int_{-\infty}^{+\infty} d\varepsilon \, u_2(\varepsilon). \qquad (A.5)$$

The only difference between $\mathcal{I}_1$ and $\mathcal{I}_2$ is that in the latter the pole $z = iv_m - \frac{i}{2\tau}$ is of order two, so the corresponding residue reads

$$\operatorname{Res}\left( u_2, iv_m - \frac{i}{2\tau} \right)$$

$$= \lim_{z \to iv_m - \frac{i}{2\tau}} \frac{d}{dz} \left\{ \left[ z - \left( iv_m - \frac{i}{2\tau} \right) \right]^2 \left[ \frac{1}{i\omega_n + iv_m - z + \frac{i}{2\tau}} \right]^2 \left[ \frac{1}{iv_m - z - \frac{i}{2\tau}} \right]^2 \right\} \qquad (A.6)$$

$$= \lim_{z \to iv_m - \frac{i}{2\tau}} (-2) \left[ \frac{1}{i\omega_n + iv_m - z + \frac{i}{2\tau}} \right]^3 (-1) = 2 \left[ \frac{1}{i\omega_n + \frac{i}{\tau}} \right]^3.$$

This gives for the integral $\mathcal{I}_2$

$$\mathcal{I}_2 = -4\pi i \frac{1}{\left( i\omega_n + \frac{i}{\tau} \right)^3}. \qquad (A.7)$$

# B    Momentum summation over the Ornstein-Zernike vertex function

We are interested in evaluating the sum

$$s_{OZ}^{m=m'} = \frac{1}{N} \sum_{\tilde{\mathbf{q}}} \frac{A}{\xi^{-2} + \tilde{q}^2}, \tag{B.1}$$

for the 1D and the 2D case. In the 1D case, we simply have, keeping in mind $\xi \gg 1$

$$s_{OZ}^{m=m',1D} = \frac{1}{2\pi} \int_{-\pi}^{+\pi} d\tilde{q}_x \frac{A}{\xi^{-2} + \tilde{q}_x^2} \xrightarrow{\xi \to \infty} \frac{1}{2\pi} 2A\xi \frac{\pi}{2} = \frac{A\xi}{2}. \tag{B.2}$$

Furthermore, by following Fig. 2 and the accompanying discussion, we note that in principle only $\tilde{\mathbf{q}}$ which retains the scattered electron-hole pair close to the Fermi surface (accounting for the smearing) should contribute to the sum. In the simplest approximation, this introduces a correction factor given by the relative size of the Fermi surface to the whole Brillouin zone. In the 1D case, there are only two momenta contributing to the Fermi surface in the whole Brillouin zone of length $2\pi$. Thus introducing a factor of $2/2\pi$ gives the final result

$$s_{OZ}^{m=m',1D} = \frac{A\xi}{2\pi}. \tag{B.3}$$

In the 2D case, the integrals are a little bit more involved

$$s_{OZ}^{m=m',2D} = \frac{1}{(2\pi)^2} \int_{-\pi}^{+\pi} d\tilde{q}_x \int_{-\pi}^{+\pi} d\tilde{q}_y \frac{A}{\xi^{-2} + \tilde{q}_x^2 + \tilde{q}_y^2}. \tag{B.4}$$

However, for $\xi \gg 1$ we can approximate the integral over the Brillouin zone with the integral over the circle of radius $\pi$, which, importantly, contains the whole Fermi surface in the half-filled case. In that case, we simply have

$$s_{OZ}^{m=m',2D} \approx \frac{1}{(2\pi)^2} \int_0^{2\pi} d\phi \int_0^{\pi} d\tilde{q} \, \tilde{q} \, \frac{A}{\xi^{-2} + \tilde{q}^2} \xrightarrow{\xi \to \infty} \frac{A}{2\pi} \ln(\pi\xi). \tag{B.5}$$

Similar to the 1D case, we should correct this result with the ratio of the length of the Fermi surface to the area of the entire Brillouin zone. In the 2D case, the situation is a little bit more involved due to the velocity contribution $v_{\mathbf{k}}^2 \sim \sin^2 k_x$, which favors momenta around $k_x \sim \pm\frac{\pi}{2}$. In the simplest approximation, we can assume that four momentum points, $\left(\pm\frac{\pi}{2}, \pm\frac{\pi}{2}\right)$, on the Fermi surface give the largest contributions, introducing a correction factor of $4/(2\pi)^2$. This is a possible source of the difference between analytical and numerical results in Fig. 4(c,d). We then have in 2D

$$s_{OZ}^{m=m',2D} \approx \frac{A\ln(\pi\xi)}{2\pi^3}. \tag{B.6}$$

To summarize, we obtained

$$s_{OZ}^{m=m'} \approx \frac{A}{2} \begin{cases} \xi/\pi, & \text{in 1D}, \\ \ln(\pi\xi)/\pi^3, & \text{in 2D}. \end{cases} \tag{B.7}$$

In 3D, there is another factor of $\tilde{q}$ in the nominator of Eq. (B.5) for spherical coordinates. Hence, even for a divergent correlation length $\xi$, there is no large $\pi$-ton contribution.

# C   Displaced Drude peak frequency and height

In order to determine the displaced Drude peak frequency, we take derivatives of Eqs. (7) and (21) with respect to frequency. For the bubble contribution, we have

$$\frac{d\sigma_{BUB}(\omega)}{d\omega} = -\frac{2\omega\tau^2\sigma_{BUB}^{dc}}{(1+\omega^2\tau^2)^2}, \tag{C.1}$$

while for the $\pi$-ton vertex contributions we get

$$\begin{aligned}\frac{d\sigma_{VERT}(\omega)}{d\omega} &= \left|\sigma_{VERT}^{dc}\right| \frac{6\omega\tau^2\left(1+\omega^2\tau^2\right)^3 - (3\omega^2\tau^2-1)6\omega\tau^2\left(1+\omega^2\tau^2\right)^2}{(1+\omega^2\tau^2)^6}\\ &= \left|\sigma_{VERT}^{dc}\right| 12\omega\tau^2\left(1+\omega^2\tau^2\right)^2 \frac{1-\omega^2\tau^2}{(1+\omega^2\tau^2)^6}.\end{aligned} \tag{C.2}$$

By adding up the two contributions, $\sigma_{TOT}(\omega) = \sigma_{BUB}(\omega) + \sigma_{VERT}(\omega)$, we obtain

$$\frac{d\sigma_{TOT}(\omega)}{d\omega} = \frac{\left|\sigma_{VERT}^{dc}\right| 12\omega\tau^2\left(1+\omega^2\tau^2\right)^2\left(1-\omega^2\tau^2\right) - 2\omega\tau^2\sigma_{BUB}^{dc}\left(1+\omega^2\tau^2\right)^4}{(1+\omega^2\tau^2)^6}. \tag{C.3}$$

The maximum is at $\frac{d\sigma_{TOT}(\omega)}{d\omega} = 0$ and given by the equation

$$6\left|\sigma_{VERT}^{dc}\right|\left(1-\omega^2\tau^2\right) - \sigma_{BUB}^{dc}\left(1+\omega^2\tau^2\right)^2 = 0. \tag{C.4}$$

We do not consider a trivial solution $\omega = 0$. By introducing $a = 6\left|\sigma_{VERT}^{dc}\right|/\sigma_{BUB}^{dc} > 1$ and $x = \omega^2\tau^2$, we may write

$$x^2 + (2+a)x + (1-a) = 0, \tag{C.5}$$

whose solutions are

$$x_{1,2} = \frac{-(2+a) \pm \sqrt{a(a+8)}}{2}. \tag{C.6}$$

We are interested only in positive $x$, so we can immediately discard the solution with the minus sign. For the solution with the plus sign to be positive, $a > 1$ needs to hold, so the criterion for the appearance of the displaced Drude peak reads $6\left|\sigma_{VERT}^{dc}\right| > \sigma_{BUB}^{dc}$. Please note that the condition $\frac{d\sigma_{TOT}(\omega)}{d\omega} = 0$ in Eq. (C.3) has either no or a single non-trivial solution. From this, it follows that the coexistence of a Drude peak and a finite-frequency maximum within our modeling is not possible. Namely, there is either a single maximum at $\omega = 0$, giving the Drude peak, or a maximum at $\omega_{DDP}$ and a minimum at $\omega = 0$, corresponding to the displaced Drude peak scenario. The frequency associated with the displaced Drude peak is then equal to

$$\omega_{DDP} = \frac{1}{\tau}\sqrt{\sqrt{3\frac{\left|\sigma_{VERT}^{dc}\right|}{\sigma_{BUB}^{dc}}\left(3\frac{\left|\sigma_{VERT}^{dc}\right|}{\sigma_{BUB}^{dc}} + 4\right)} - \left(1 + 3\frac{\left|\sigma_{VERT}^{dc}\right|}{\sigma_{BUB}^{dc}}\right)}, \tag{C.7}$$

which in the limit $6\left|\sigma_{VERT}^{dc}\right| \gg \sigma_{BUB}^{dc}$ reduces to $\omega_{DDP} = \tau^{-1}$. In the latter case, the height of the displaced Drude peak equals

$$\sigma_{TOT}(\omega_{DDP}) = \sigma_{BUB}^{dc} - \frac{1}{4}\left|\sigma_{VERT}^{dc}\right|. \tag{C.8}$$

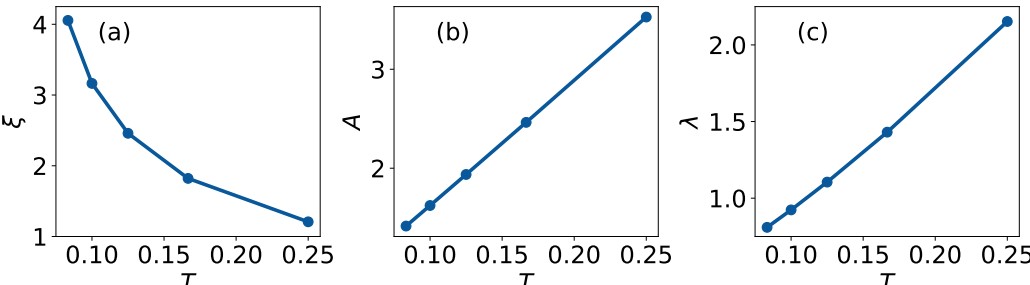

Fig. 7: Temperature dependence of the 2D Ornstein-Zernike parameters $\xi$, $A$, and $\lambda$ from Ref. [31], extracted by fitting the Ornstein-Zernike form in Eq. (9) to the RPA $\pi$-ton vertex function.

## D  Temperature dependence of the Ornstein-Zernike parameters

In Fig. 7, we show the temperature dependence of the Ornstein-Zernike parameters $\xi$, $A$, and $\lambda$ extracted in Ref. [31] by fitting the Ornstein-Zernike vertex form to the RPA $\pi$-ton vertex function for the case with the fermion lifetime $\tau^{-1} = 0.1547 + 1.637\, T^2$ in the 2D case. The corresponding temperature dependencies respectively read

$$\xi = \frac{0.30}{T} + 10^{-3} \exp^{\frac{0.51}{T}}, \quad A = 0.41 + 13T^{1.03}, \quad \text{and} \quad \lambda = 0.38 + 10.6T^{1.29}. \tag{D.1}$$

Note that although the correlation length is extracted from the RPA $\pi$-ton vertex function showing a finite transition temperature at $T_c \approx 1/19$, it is fitted with an exponential function, resembling the true 2D exponential divergence of the correlation length as zero temperature is approached.

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
