# Peer review of "Analytical expression for $\pi$-ton vertex contributions to the optical conductivity"

_SciPost Physics, doi:SciPost Phys. 18, 138 (2025)_

## Round 1 · Referee Report · Simone Fratini (Referee 1) · 2024-11-15

Report

This paper addresses theoretically one of the possible microscopic scenarios leading to displaced Drude peaks (DDP), a widespread experimental phenomenon occurring in a variety of quantum materials. The focus here is on a specific class of vertex corrections arising in the particle-hole channel, that have previously been shown by the authors to affect the optical conductivity beyond the common "bubble" approximation.

Numerous examples have been analyzed in recent times by the authors as well as by other groups, illustrating the realization of this general mechanism in different microscopic models and dimensionalities. The present long paper rationalizes these observations, setting up an approximate theory that: (i) provides a unified framework to interpret the existing data; (ii) serves as a useful parametrization of the numerical results, including a formula (Eq.21) that could be applied directly to the interpretation of experiments; (iii) provides indications on how to tweak the microscopic parameters in order to match the experimental observations.

The paper is written in a very pedagogical format, that is well suited for this journal. The topic itself also nicely matches the recent trend set by related DDP studies also published in SciPost.

Although I strongly recommend publication of this work, I have some general and few more specific questions:

  1. Are the vertex corrections addressed by the authors in any way related to Bang and Kotliar's corrections to the optical conductivity found via the slave boson technique [PRB 48 9898 (1993)], originating there from the incoherent motion of the charge carriers and their coupling to low-energy spin excitations?

  2. Eqs. 1-2 bear some similarity with the starting point of the phenomenological theory of strange metals given by S. Caprara and collaborators [Communications Physics 5 (1), 10 (2022)]. Can the authors show whether and how the pi-tons would affect the temperature dependence of the resistivity (i.e. illustrate the impact of Eq. 22 on an otherwise T^2-dependence)? Or does the Fermi liquid assumption underlying their approach automatically lead to a Fermi liquid-like behavior in transport?

  3. l.145, even though it has appeared elsewhere, a figure depicting the corresponding Feynman diagrams could be useful here

  4. Based on Eq. 21, is it possible to obtain an anomalous frequency decay of the Drude peak in the optical conductivity? In particular, could the authors comment if this can be made to match the shape that is often seen in experiments when DDPs are absent, and that was phenomenologically analyzed for example in [Nature Communications 14 (1), 3033 (2023)]?

  5. In Eq. 22, the pi-ton contribution always suppresses the d.c. conductivity, while in other references cited in this mannuscript it was found that pi-tons can either suppress or enhance the conductivity depending on the temperature range. Could the authors comment on this?

  6. l.290 is it obvious that the same parametric form of the lifetime assumed here should hold both in 1D and 2D?

  7. l.362 this should probably read "at the beginning of Section 3.2"

  8. l.369 what is the extra microscopic ingredient used in Ref. 32 that is missed here, and why should it be more correct than the original assumption Eq. 25, therefore motivating a modification of the temperature behavior of the correlation length?

  9. From the comparison of the analytical Eqs. 7 and 21, is it possible to derive precise conditions for the emergence of a DDP? And for a possible coexistence of a Drude peak together with a finite-frequency maximum?

Recommendation

Publish (easily meets expectations and criteria for this Journal; among top 50%)

  • validity: high
  • significance: high
  • originality: high
  • clarity: top
  • formatting: perfect
  • grammar: perfect

Author:  Anna Kauch  on 2025-03-14  [id 5290]

(in reply to Report 1 by Simone Fratini on 2024-11-15)

\referee{This paper addresses theoretically one of the possible microscopic scenarios leading to displaced Drude peaks (DDP), a widespread experimental phenomenon occurring in a variety of quantum materials. The focus here is on a specific class of vertex corrections arising in the particle-hole channel, that have previously been shown by the authors to affect the optical conductivity beyond the common "bubble" approximation.

Numerous examples have been analyzed in recent times by the authors as well as by other groups, illustrating the realization of this general mechanism in different microscopic models and dimensionalities. The present long paper rationalizes these observations, setting up an approximate theory that: (i) provides a unified framework to interpret the existing data; (ii) serves as a useful parametrization of the numerical results, including a formula (Eq.21) that could be applied directly to the interpretation of experiments; (iii) provides indications on how to tweak the microscopic parameters in order to match the experimental observations.

The paper is written in a very pedagogical format, that is well suited for this journal. The topic itself also nicely matches the recent trend set by related DDP studies also published in SciPost.

Although I strongly recommend publication of this work, I have some general and few more specific questions:}

\reply{We sincerely thank the Referee for reviewing the manuscript, for the positive evaluation of our work, and for recommending its publication. Below, we address the Referee's questions, which have helped us improve the manuscript}.

\referee{1. Are the vertex corrections addressed by the authors in any way related to Bang and Kotliar's corrections to the optical conductivity found via the slave boson technique [PRB 48 9898 (1993)], originating there from the incoherent motion of the charge carriers and their coupling to low-energy spin excitations?}

\reply{This is an interesting question. There are several theories with some form of single-boson exchange contribution to the optical conductivity as also in the work by Bang and Kotliar. Indeed, the (RPA) $\pi$-ton vertex function, depicted in the newly added Fig.~1(b), can be approximated by the Ornstein-Zernike form in the vicinity of the antiferromagnetic phase transition. In this case, the entire ladder vertex in the transversal particle-hole channel in Fig. 1(b) can be replaced by a single boson. Consequently, the $\pi$-ton vertex contribution diagram in Fig.~1(a) shares the same topology as the rightmost diagram in Fig.~1(c) of PRB \textbf{48}, 9898 (1993).

However, the physics is different. In our case we have strong antiferromagnetic spin fluctuations with a long antiferromagnetic correlation length. This is the very essence of the $\pi$-ton vertex corrections. To the best of our understanding, the slave boson or large $N$ expansion, instead describes physics akin to resonant valence bond (RVB) states. This gives rise to a very different contribution to the optical conductivity: the vertex correction in the paper by Bang and Kotliar does not give a displaced Drude peak but rather a rescaling of the bubble contribution.

Besides this general difference, there is of course also the difference that our work focuses on weakly correlated metallic systems, while the $t$-$J$ model is instead well-suited for very strong electronic correlations. In our case, we deal with slightly renormalized, coherent Fermi liquid-like quasiparticles. The spin fluctuations are overdamped oscillators with a mass determined by the (large) correlation length, and the coupling constant $g$ is featureless and contained within the Ornstein-Zernike parameter $A$. In contrast, for example, PRB \textbf{48}, 9898 (1993) highlights that the corresponding fermion-boson vertices develop strong momentum dependence.

To summarize, although we effectively consider a topologically similar vertex correction to that in PRB \textbf{48}, 9898 (1993), the physics and resulting optical conductivity are very different. Nonetheless, we know cite the work by Kotliar {\it et al.} in our paper.}

\referee{2. Eqs. 1-2 bear some similarity with the starting point of the phenomenological theory of strange metals given by S. Caprara and collaborators [Communications Physics 5 (1), 10 (2022)]. Can the authors show whether and how the pi-tons would affect the temperature dependence of the resistivity (i.e. illustrate the impact of Eq. 22 on an otherwise T\^2-dependence)? Or does the Fermi liquid assumption underlying their approach automatically lead to a Fermi liquid-like behavior in transport?}

\reply{This is an intriguing question; however, it does not have a completely unambiguous answer. To address it, let us start by considering the simpler case of the bubble contribution $\sigma_{BUB}^{dc} = 2g(\varepsilon_{F})\left\langle v^2\right\rangle_{FS}\tau$. Assuming that $g(\varepsilon_{F})$ and $\left\langle v^2\right\rangle_{FS}$ have weak temperature dependencies, the temperature dependence of the resistivity is governed by the scattering rate $\rho_{BUB}^{dc} \sim \tau^{-1}$. With the Fermi liquid-like assumption of $\tau^{-1}\sim a+bT^2$ this then leads to the Fermi liquid-like behavior in transport. We also added a constant term $a$ to account for static impurities or some other source of scattering.

The situation with the $\pi$-ton vertex contribution is a bit more involved. For instance, in 2D Eq.~(22) suggests $\sigma_{VERT}^{dc} = -A\;T\tau^2\sigma_{BUB}^{dc}\ln(\pi\xi)/\pi^3$. Now, the temperature dependence arises not only from the scattering rate $\tau$ but also from the correlation length $\xi$ and parameter $A$. Their temperature dependencies can vary across different systems, even though the scattering rate remains Fermi liquid-like. Nevertheless, we can express the full resistivity with the $\pi$-ton contribution included as $\rho_{TOT}=\sigma^{-1}_{TOT}=\frac{1}{\sigma_{BUB}^{dc}\left(1 - A\;T\tau^2\ln(\pi\xi)/\pi^3\right)}\sim \frac{\tau^{-1}}{\left(1 - A\;T\tau^2\ln(\pi\xi)/\pi^3\right)}$.

To proceed, we consider an ideal 2D system as an example, where $\xi = p\exp^\frac{r}{T}$, and assume a linear-$T$ dependence for the parameter $A\approx c + dT$, as suggested by Fig.~6(b). Together with the Fermi liquid-like scattering rate, this then gives
$$\rho_{TOT}(T)\sim\frac{a+bT^2}{1-\left(c + dT\right)(a+bT^2)^{-2}\left(r+T\ln p\pi\right)/\pi^3}\;.$$ For very high $T\to\infty $ temperatures, the resistivity then also exhibits Fermi liquid-like behavior since the denominator approaches 1. In this particular case, the Fermi liquid-like assumption for the scattering rate leads also to a Fermi liquid-like behavior in transport even with the $\pi$-ton vertex contributions included. More generally, we have $\rho_{TOT}(T\to\infty)\sim\tau^{-1}$. Interestingly, however, at very low temperatures ($T \to 0$), the resistivity becomes linear in $T$, despite the scattering rate exhibiting Fermi liquid-like behavior. This is most easily seen from the above equation by noting the linear in $T$ terms in the denominator, $\rho_{TOT}(T\to 0)\to\frac{a+bT^2}{C - DT}$, where $C=1-\frac{cr}{a^2\pi^3}$ and $D=\frac{dr+c\ln p\pi}{a^2\pi^3}$ are temperature-independent constants, coming from $\tau$, $A$ and $\xi$. To ensure a physical solution, $C-DT>0$ needs to hold for each $T$. The Taylor expansion for $T\to 0$ now gives $\rho_{TOT}(T\to 0)\approx\frac{a}{C}+\frac{aD}{C^2}T+\frac{b}{C}T^2 + ...\;$, suggesting that the linear in $T$ term becomes leading at low enough temperatures, together with the renormalization of the residual resistivity $a\to\frac{a}{C}$.

However, please keep in mind that this analysis applies to an ideal 2D system. Different temperature dependencies of $A$ and $\xi$ could lead to alternative temperature dependencies of the resistivity. Additionally, in Communications Physics 5(1), 10 (2022), only the bubble contribution with renormalized fermionic lines was considered. In contrast, our approach to optical conductivity includes vertex contributions beyond the bubble term.

We would like to sincerely thank the Referee for bringing up this discussion. We believe this result further enhances the quality and significance of our manuscript. Therefore, we have added a whole new Subsection 3.2.3 where we discuss the dc resistivity in the presence of $\pi$-tons and a linear $T$ behavior for low temperatures in the ideal 2D case.We also comment on this result in Abstract and Conclusion.}

\referee{3. l.145, even though it has appeared elsewhere, a figure depicting the corresponding Feynman diagrams could be useful here}

\reply{As recommended by the Referee, we have included a figure illustrating the corresponding Feynman diagrams.}

\referee{4. Based on Eq. 21, is it possible to obtain an anomalous frequency decay of the Drude peak in the optical conductivity? In particular, could the authors comment if this can be made to match the shape that is often seen in experiments when DDPs are absent, and that was phenomenologically analyzed for example in [Nature Communications 14 (1), 3033 (2023)]?}

The $\pi$-ton vertex corrections fall off faster than the bubble contribution for large frequencies. Hence, there cannot be an anomalous frequency decay in the sense of [Nature Communications 14 (1), 3033 (2023)]. We think that paper is right to
associate this behavior with the frequency dependence of the self-energy (effective mass) within the bubble contribution.

\referee{5. In Eq. 22, the pi-ton contribution always suppresses the d.c. conductivity, while in other references cited in this manuscript it was found that pi-tons can either suppress or enhance the conductivity depending on the temperature range. Could the authors comment on this?}

\reply{We assume that the Referee is referring to Refs.~[31] and [32]. Starting with Ref.~[32], a careful analysis of numerical convergence and adaptive integration demonstrated that, as the phase transition is approached and the $\pi$-ton vertex function is accurately described by the Ornstein-Zernike form, the $\pi$-ton vertex contribution consistently suppresses the dc conductivity. This finding is further reinforced analytically in this manuscript. On the other hand, Ref.~[31] argued that, far from the phase transition at very high temperatures, the effect could be reversed, with the $\pi$-ton contributions enhancing the dc conductivity. Therefore, based on these two results, one might suggest—albeit without a precise quantitative estimate—that $\pi$-tons suppress the dc conductivity near the transition, while far from the transition, they enhance it. However, the latter effect requires more careful analysis. In this regard, it is worth noting that Eq.~(22) indeed suggests that in the 2D case for $\xi \ll 1$ the sign of $\pi$-ton vertex contributions becomes positive, which is somewhat in line with the results of Ref.~[31]. We added the corresponding comment at the beginning of Sec.~3.}

\referee{6. l.290 is it obvious that the same parametric form of the lifetime assumed here should hold both in 1D and 2D?}

No this is not obvious. We here assumed a Fermi liquid form with a finite residual scattering, which is quite generic. For 1D and 2D systems there can be enhanced scattering at low temperatures at e.g.\ spin fluctuations. The exact form of this scattering is still an open debate.
The used Fermi liquid form including the residual scattering for $T\rightarrow 0$ has been obtained by fitting to 2D parquet D$\Gamma$A results for the half-filled Hubbard model. However, given the restricted temperature range of that fit, we do not want to claim the aforementioned debate to be settled nor that 1D has the same behavior. We
improved our presentation of this point.

\referee{7. l.362 this should probably read "at the beginning of Section 3.2"}

\reply{We have made the change as suggested by the Referee.}

\referee{8. l.369 what is the extra microscopic ingredient used in Ref. 32 that is missed here, and why should it be more correct than the original assumption Eq. 25, therefore motivating a modification of the temperature behavior of the correlation length?}

\reply{In Ref.~[32], the $\pi$-ton vertex function (proportional to the magnetic susceptibility) was calculated within the RPA. This RPA $\pi$-ton vertex function (magnetic susceptibility) exhibits a divergence at a finite temperature $T_c$, signaling a finite-temperature phase transition, with the correlation length algebraically diverging as $\xi \sim (T-T_c)^{-\nu}$. This scenario is depicted in Figs.~4(g) and (h), showing consistent results with those in Ref.~[32].

In an ideal 2D system, however, magnetic order should not occur at a finite temperature; its appearance in Ref.~[32] was simply a consequence of employing the RPA. To address this issue, the behavior of the correlation length was modified to $\xi \sim \exp^\frac{1}{T}$ (this term wins over the algebraic terms at low enough temperatures). This form aligns with the fact that the transition should occur only at zero temperature and accurately captures the exponential growth of the correlation length as the transition is approached, consistent with the behavior expected in an ideal 2D system.

In this context, the scenario with $\xi \sim \exp^\frac{1}{T}$ is more suitable for investigating $\pi$-ton effects in ideal 2D systems. However, one can also consider a quasi-2D system where a phase transition occurs at a finite temperature. To address this, we present the description of $\pi$-ton effects for both scenarios.}

\referee{9. From the comparison of the analytical Eqs. 7 and 21, is it possible to derive precise conditions for the emergence of a DDP? And for a possible coexistence of a Drude peak together with a finite-frequency maximum?}

\reply{Please note that in Appendix C, we indeed derived the precise condition for the emergence of a DDP, given by $6\left|\sigma_{VERT}^{dc}\right|/\sigma_{BUB}^{dc}>1$, as well as the DDP frequency $\omega_{DDP}$ (see Eq.~(C.7)). It is also discussed in the main text at the beginning of Sec.~3.

From the condition $\frac{d\sigma_{TOT}(\omega)}{d\omega}=0$ (see Eq.~(C.3))}, it also follows that the coexistence of a Drude peak and a finite-frequency maximum within our modeling is not possible. Namely, there is either a single maximum at $\omega=0$, giving the Drude peak, or a maximum at $\omega_{DDP}$ and a minimum at $\omega=0$, corresponding to the DDP scenario. We now also explicitly state this in Appendix C. Please note that this can change in the case of a frequency-dependent self-energy, such as that in dynamical mean-field theory, when already at the bubble contribution level a Drude peak and a finite frequency maximum can be obtained.

---

## Round 1 · Referee Report · Anonymous (Referee 2) · 2024-11-20

Report

The paper analytically evaluates the pi-ton vertex corrections to optical conductivity and discusses the implications of those corrections for the displaced Drude peak phenomenology. Earlier work by some of the authors has established the existence of pi-ton vertex corrections, the novel idea of this work is to introduce additional approximations that allow for analytical evaluation.

Overall, the topic of the investigation is interesting and progress obtained from a simplified evaluation of momentum integrals significant, but I feel there are parts where the discussion could be improved.

Concerning the derivation, I do not understand the separation into hole and electron parts well. Paper says that taking w_n > 0 implies a restriction concerning which fermionic frequencies contribute. Is the statement that contributions from other frequencies vanishes strictly or is this an approximation? If the latter, is this approximation necessary, is it better at lower temperatures? What does this imply for the vanishing bosonic Matsubra frequency?

To me it is insufficiently clear what does "adaptive integration" result really mean. Is Ornstein Zernike form for the susceptibility assumed also there and the analytical approximation refers mainly to simplification of momentum integrals? A reader could benefit by more explicit discussion of where approximations are.

Given the simple forms assumed for the Green's function and vertex in analytics, could one write the expressions for them on the real frequency axis and perform the integrals there?

Concerning the physics, the paper gives a criterion for the occurrence of the Drude peak in terms of that the vertex correction needs to exceed a given threshold. On the other hand, in the low T limit it also gives unphysical negative conductivity. What breaks down? Do not all approximations introduced by the authors become better at low temperature?

Concerning the Ward identity sum-rule, perhaps one could explicitly write that the vanishing of zero-Matsubara frequency value is consistent with the absence of momentum dependence of self-energy. Which aspect of the vertex function in the approximation guarantees that sigma_vert(w) indeed behaves in this way?

Authors consider 1d and 2d. What happens in 3d? Some of the materials that display displaced Drude peak phenomenology actually exist in 3d.

Recommendation

Ask for minor revision

  • validity: -
  • significance: -
  • originality: -
  • clarity: -
  • formatting: -
  • grammar: -

Author:  Anna Kauch  on 2025-03-14  [id 5291]

(in reply to Report 2 on 2024-11-20)

\reply{First of all, we would like to sincerely thank the Referee for reviewing our manuscript, for the time spent, and especially for the criticism raised that helped us improve our manuscript.}

\referee{The paper analytically evaluates the pi-ton vertex corrections to optical conductivity and discusses the implications of those corrections for the displaced Drude peak phenomenology. Earlier work by some of the authors has established the existence of pi-ton vertex corrections, the novel idea of this work is to introduce additional approximations that allow for analytical evaluation. Overall, the topic of the investigation is interesting and progress obtained from a simplified evaluation of momentum integrals significant, but I feel there are parts where the discussion could be improved.}

\reply{We are glad to hear that the Referee finds our results interesting and significant. In response to the Referee's comment suggesting that there are some parts of the manuscript where the discussion can be improved, we revised the manuscript by addressing each of the Referee's questions point by point.}

\referee{Concerning the derivation, I do not understand the separation into hole and electron parts well. Paper says that taking w\_n $>$ 0 implies a restriction concerning which fermionic frequencies contribute. Is the statement that contributions from other frequencies vanishes strictly or is this an approximation? If the latter, is this approximation necessary, is it better at lower temperatures? What does this imply for the vanishing bosonic Matsubara frequency?}

\reply{
At the light-electron interaction vertex, an electron at frequency $\nu$ is annihilated
and one at frequency $\nu+\omega$ is created, resulting in the propagation of a particle and a hole. We here only analytically continue this to Matsubara frequencies.

Our approach closely follows the derivation of the bubble contribution as presented in Ref.~[42]. There, $\omega_n>0$ is also assumed from the very beginning, which corresponds to $\omega>0$ on the real axis.

Mathematically, $G$ in Eq.~(1) can be expressed in terms of the introduced $G^{e/h}$ exactly as $G = G^{e} + G^{h}$, where $G^{e}$ and $G^{h}$ have poles in opposite halves of the complex plane. This implies that for the bubble contribution, we have $\chi_{BUB}(i\omega_n)\propto \left[G^e(\mathbf{k},i\nu_m)+G^h(\mathbf{k},i\nu_m)\right]\left[G^e(\mathbf{k},i\nu_m+i\omega_n)+G^h(\mathbf{k},i\nu_m+i\omega_n)\right]$, resulting in four contributions, each containing a product of two Green's functions.

These products only yield a non-zero contribution to $\chi_{BUB}$, if the two Green's functions have poles in opposite halves of the complex plane. For $\omega_n>0$, this condition is satisfied only when $\nu_m<0$ and the only non-vanishing contribution is $G^h(\mathbf{k},i\nu_m)G^e(\mathbf{k},i\nu_m+i\omega_n)$, while the other three products contribute zero.
%($G^h(\mathbf{k},i\nu_m)G^h(\mathbf{k},i\nu_m+i\omega_n)$ and $G^e(\mathbf{k},i\nu_m)G^e(\mathbf{k},i\nu_m+i\omega_n)$ have all their poles located in the same half of the complex plane, while $G^e(\mathbf{k},i\nu_m)G^h(\mathbf{k},i\nu_m+i\omega_n)$ cannot be realized for $\omega_n>0$).
The derivation is exact, with the $\omega_n=0$ case being understood as the $\omega_n\to0^+$ limit. Further, a restriction to $\omega_n>0$ is possible since the bosonic optical conductivity is symmetric in $\omega$.

To provide greater clarity on these points in the manuscript, we have moved the introduction of $G^{e/h}$ to Subsection 2.1 during the evaluation of the bubble contribution and revised the corresponding discussion.}

\referee{To me it is insufficiently clear what does "adaptive integration" result really mean. Is Ornstein Zernike form for the susceptibility assumed also there and the analytical approximation refers mainly to simplification of momentum integrals? A reader could benefit by more explicit discussion of where approximations are.}

\reply{We have used adaptive integration to evaluate the full expression given in Eq.~(8) on the real frequency axis with (i) the Ornstein-Zernike form of the vertex function and (ii) Fermi liquid-like Green's functions given in Eq.~(1). We have used the somewhat lengthy expressions for Eq.~(8) on the real frequency axis derived in Ref.~[31] and also repeated in Appendix A of Ref.~[32]. No other approximations than (i) and (ii) were used. We have now extended the introductory part of Section 3.1 to make this point clear. The results from numerical adaptive integration are then compared in Section 3.1 to the analytical expressions, i.e., Eqs.~(20) and~(21). The analytical expressions were derived with additional approximations, as presented in detail in Section 2.2.}

\referee{Given the simple forms assumed for the Green's function and vertex in analytics, could one write the expressions for them on the real frequency axis and perform the integrals there?}

\reply{That is exactly what we did and called it ``adaptive integration" result, because adaptive integration was necessary to converge the multidimensional integrals in momentum and real frequency space. The expressions on the real frequency axis were already written down in Refs.~[31] and~[32] and we used them. As already mentioned, we explain it now in more detail at the beginning of Section 3.1.}

\referee{Concerning the physics, the paper gives a criterion for the occurrence of the Drude peak in terms of that the vertex correction needs to exceed a given threshold. On the other hand, in the low T limit it also gives unphysical negative conductivity. What breaks down? Do not all approximations introduced by the authors become better at low temperature?}

To keep things simple and arrive at analytical expressions, we did not include feedback
effects, namely: (i) how the vertex corrections modify (dampen) the self-energy and
thus self-consistently change (weaken) the $\pi$-ton contribution and (ii) how large vertex corrections in the particle-hole transversal vertex also impact the particle-hole and in particular the particle-particle channel. Both effects are included in the numerical parquet calculations that lead to the obeservation of $\pi$-tons in the first place, but not in the (semi)analytical calculations of the present paper. We now emphasize this in the manuscript.

\referee{Concerning the Ward identity sum-rule, perhaps one could explicitly write that the vanishing of zero-Matsubara frequency value is consistent with the absence of momentum dependence of self-energy. Which aspect of the vertex function in the approximation guarantees that sigma\_vert(w) indeed behaves in this way?}

\reply{That is a good and important point. We have now added a comment to the manuscript clarifying that we consider a momentum-independent self-energy, for which the entire optical spectral weight is fully accounted for by the bubble contribution.

Determining which specific aspect of the vertex function ensures the vanishing of the optical spectral weight is not straightforward. For instance, the RPA $\pi$-ton vertex function employed in Refs.~[31, 32] results in a non-vanishing spectral weight, even with a momentum-independent self-energy. In contrast, the Ornstein-Zernike $\pi$-ton vertex function only appears to redistribute the spectral weight. However, it is important to note that, for the sake of simplifying the analytic derivation, several approximations were made when obtaining the final expression for the $\pi$-ton contribution to the optical conductivity. Consequently, it is unclear whether this behavior is an intrinsic property of the Ornstein-Zernike $\pi$-ton vertex function or a result of the specific approach used to derive the overall vertex contribution.}

\referee{Authors consider 1d and 2d. What happens in 3d? Some of the materials that display displaced Drude peak phenomenology actually exist in 3d.}

The higher the dimension the weaker the $\pi$-ton vertex correction becomes. Even for a divergent correlation length, we anticipate no large $\pi$-ton vertex correction in 3D.
We thus do not expect a displaced Drude peak from this mechanism in 3D. We now mention this explicitly in the Conclusion, Section 2, and Appendix B.

---

## Round 2 · Referee Report · Simone Fratini (Referee 1) · 2025-3-21

Report

I am satisfied with the changes, the paper can be published in the present form.

Recommendation

Publish (surpasses expectations and criteria for this Journal; among top 10%)

---

## Round 2 · Referee Report · Anonymous (Referee 2) · 2025-4-1

Report

The authors have addressed the raised issues satisfactorily. I recommend the paper for publication.

Recommendation

Publish (easily meets expectations and criteria for this Journal; among top 50%)

---

## Round 2 · Author Response

We sincerely thank both Referees for their positive and constructive feedback on our manuscript.

In response to their comments and suggestions, we have revised the manuscript as detailed below.

We believe these updates have improved the importance and quality of the manuscript.

---

## Round 2 · List of Changes

We have made the following revisions to the manuscript:

  • Added a new Subsection 3.2.3 discussing the temperature dependence of the DC resistivity in the ideal 2D case. Corresponding comments have also been included in the Abstract and Conclusion.

  • Introduced Fig. 1, which diagrammatically represents the current-current correlation function with the $\pi$-ton vertex corrections considered in the manuscript.

  • Moved the introduction of the Green’s function $G^{e/h}$ to the section where the bubble contribution is evaluated and provided a more detailed explanation of the evaluation process.

  • Included a footnote on single-boson vertex corrections within the slave boson theory, along with relevant references. Valuable discussions with M. Grilli are acknowledged.

  • Added comments on $\pi$-ton vertex corrections in 3D and explained why we assume them to be small.

  • Briefly discussed the potential behavior of $\pi$-tons far from the transition, referring to previous works.

  • Explained why considered $\pi$-ton vertex corrections may lead to unphysical negative conductivity.

  • Emphasized that the optical sum spectral weight is entirely determined by the bubble contribution due to the considered momentum-independent self-energy.

  • Clarified the meaning of the adaptive integration results.

  • Provided arguments explaining why our approach does not allow for a Drude peak alongside a finite-frequency maximum, but rather results in either a Drude peak with no additional finite-frequency maximum or a displaced Drude peak.

  • Made additional minor revisions in response to the Referees' reports.

---

## Editorial Decision

published